# Synchronization between peripheral circadian clock and feeding-fasting cycles in microfluidic device sustains oscillatory pattern of transcriptome

Onelia Gagliano [1,2,3], Camilla Luni [4,5], Yan Li[3], Silvia Angiolillo[1,2], Wei Qin[1,2], Francesco Panariello [6,7], Davide Cacchiarelli[6,7], Joseph S. Takahashi [3,8✉] & Nicola Elvassore[1,2,9✉]

The circadian system cyclically regulates many physiological and behavioral processes within the day. Desynchronization between physiological and behavioral rhythms increases the risk of developing some, including metabolic, disorders. Here we investigate how the oscillatory nature of metabolic signals, resembling feeding-fasting cycles, sustains the cell-autonomous clock in peripheral tissues. By controlling the timing, period and frequency of glucose and insulin signals via microfluidics, we find a strong effect on *Per2*::Luc fibroblasts entrainment. We show that the circadian *Per2* expression is better sustained via a 24 h period and 12 h:12 h frequency-encoded metabolic stimulation applied for 3 daily cycles, aligned to the cell-autonomous clock, entraining the expression of hundreds of genes mostly belonging to circadian rhythms and cell cycle pathways. On the contrary misaligned feeding-fasting cycles synchronize and amplify the expression of extracellular matrix-associated genes, aligned during the light phase. This study underlines the role of the synchronicity between life-style-associated metabolic signals and peripheral clocks on the circadian entrainment.

[1] Department of Industrial Engineering (DII), University of Padova, Padova, Italy. [2] Venetian Institute of Molecular Medicine (VIMM), Padova, Italy. [3] Department of Neuroscience, Peter O'Donnell Jr. Brain Institute, University of Texas Southwestern Medical Center, Dallas, TX, USA. [4] Shanghai Institute for Advanced Immunochemical Studies (SIAIS), ShanghaiTech University, Shanghai, China. [5] Department of Civil, Chemical, Environmental and Materials Engineering (DICAM), University of Bologna, Bologna, Italy. [6] Telethon Institute of Genetics and Medicine (TIGEM), Pozzuoli, Italy. [7] Department of Translational Medicine, University of Naples "Federico II", Naples, Italy. [8] Howard Hughes Medical Institute, University of Texas Southwestern Medical Center, Dallas, TX, USA. [9] Stem Cell and Regenerative Medicine Section, University College London GOS Institute of Child Health, London, UK. ✉email: Joseph.Takahashi@utsouthwestern.edu; n.elvassore@ucl.ac.uk

Many processes of mammalian behavior and physiology, such as sleeping and feeding, are cyclically regulated during the 24 h solar day by the circadian clock system. The molecular mechanism governing circadian rhythmicity is based on a complex program of several interconnected transcriptional and post-translational negative feedback loops where the major players, the two positive regulators, CLOCK and BMAL1, and the repressors belonging to the *Cryptochrome* (CRY1 and CRY2) and *Period* (PER1, PER2, PER3) families, in turn, lead to circadian oscillations[1–3]. The circadian input consists of two major pathways. The first is the light input pathway via the hypothalamic suprachiasmatic nuclei (SCN), known as the circadian pacemaker. The second is the feeding input pathway: temporal feeding restriction changes the phase of circadian gene expression in peripheral tissues[4–6], which is highly conserved among plants, different animal lineages, and including human[7,8].

Chronic desynchronization between physiological and behavioral rhythms (as in shift work, sleep disruption, or abnormal feeding schedules in humans) carries a significant risk of diverse disorders, ranging from sleep disorders to diabetes, obesity, cardiovascular disease, and cancer[9–13]. In particular, chronic desynchronization of food intake from the circadian clock system in humans is an important factor in the onset of metabolic diseases[9,15]. Thus, a deeper understanding of the mechanisms by which environmental inputs affect the circadian clock and the rhythmicity of cellular functions is critically important for the prevention of diseases.

In the past, peripheral clocks were thought to be synchronized primarily by the hypothalamic suprachiasmatic nuclei SCN in response to light[16]. However, in the last decades, it has become clear that behavioral rhythmicity is one of the major entrainment cues in terms of synchronization of peripheral clocks[5,17–19]. Specifically, it was shown that rhythmic feeding and caloric content are involved in various nutrient-sensing pathways, such as insulin/IGF-1, SIRT1, NAMPT, AMPK, PGC-1a, mTOR, GSK3b, and FGF21, that are both necessary and sufficient to synchronize the circadian expression of peripheral clocks[20–26].

Although there is a lot of scientific evidence about the link between the circadian clock and metabolism, the design of an experimental strategy for investigating and dissecting the contribution of specific oscillatory metabolic patterns on the circadian clock is a challenge. In vivo studies provide information about the entire organism, with limited capability of dissecting the contributions of the different circadian inputs (such as light, feeding/fasting regimes, activity, temperature) or other major factors (such as the endocrine system, metabolite fluctuations) in the circadian system of peripheral tissues[27–29]. On the other hand, in vitro circadian experiments[30–32], based on a single pulse of metabolic stimulation mimicking the onset of the feeding phase, showed evidence of peripheral circadian alteration[4,33–36]. However, these in vitro experiments show limited capability in reproducing the rhythms of metabolic signals associated with dietary regimens and timing.

In this work, we start to rationally investigate how the dynamic oscillatory nature of metabolic signals, resembling the daily feeding-fasting cycle, alter or, even profoundly reset, the cell-autonomous circadian clock in peripheral tissues. We hypothesize that frequency-encoded metabolic stimulations affect the cell-autonomous circadian clock and alter the rhythmicity of the cellular transcriptome. We also explore whether mismatch of cell-autonomous circadian clock and phases of 24 h metabolic cycles could affect circadian rhythms.

We take advantage of microfluidic technology to perform periodic cyclic stimulations[37,38] under controlled microenvironment conditions. The characteristic micrometric dimension of microfluidics supports laminar flow, which results in highly reproducible mass transport condition, accurate spatio-temporal control of the concentration of solute over a cell monolayer, and fast change of medium volume, typically of few microliters. Microfluidics can be easily automatized for remote control of culture conditions and enables multiplexing analyses. In the circadian field, these microfluidics properties have been exploited for long-term bacteria and algae measurements of synchronized oscillations in a chemostatic environment by continuous perfusion[39–44]. Recent studies reported microfluidic approaches for studying the molecular mechanisms of circadian gene oscillations in mammalian cells by period pulsed stimulations[45].

Here, we specifically develop a microfluidic approach to perform periodic cyclic stimulations[37,38] under controlled microenvironmental conditions, while continuously monitoring circadian oscillations. Long-term circadian study of mammalian microfluidic cell culture shows the unexpected importance of frequency-encoded metabolic perturbations for resetting the circadian clock. Furthermore, through cyclic temporal stimulations, we dissect the contributions of the feeding and fasting phases on clock resetting with oscillatory stimulation synchronous and asynchronous with the cell-autonomous circadian clock.

## Results

**Microfluidic technology for circadian studies.** We designed and developed a microfluidic platform (Supplementary Fig. S7) to perform frequency-encoded metabolic perturbations, while recording circadian gene expression by luminescence detectors (Fig. 1a). This microfluidic setup was connected with an external liquid handling system for accurate control of medium delivery (Fig. 1b). The recording of luminescence signals from the two 1.75 µL microfluidic cell culture chambers was performed under a temperature-controlled microscope providing a spatial resolution of 20 µm.

As proof of concept of circadian synchronization of mammalian cell cultures in a microfluidic environment, Fig. 1c and Supplementary Movie 1 show a time-lapse of luminescence signal from mouse *Per2*::Luc fibroblasts. The microfluidic experimental setup permits recording of highly reproducible, robust, and spatially resolved oscillatory behavior of the cell population after dexamethasone shock (Fig. 1d). To further verify the feasibility of microfluidic circadian investigation, we performed a qPCR analysis of *Per2* and *Bmal1* gene expression in Human Foreskin Fibroblasts (HFF) for 48 h after dexamethasone shock (Fig. 1e), a typical protocol for in vitro circadian entrainment[46]. The expression of the two genes was rhythmic and in anti-phase of 12 h, according to the mechanisms of transcription/translation feedback loops that regulate the circadian clock at the molecular level. These results showed the feasibility of circadian studies using microfluidic technology.

The capability to manipulate and control the microfluidic cellular microenvironment by external liquid handling offers unlimited strategies of how cell culture can be perfused, from low to high frequency of periodic perfusion[47]. However, how the perfusion frequency affects the circadian clock needs to be investigated, in consideration of a balance between nutrient delivery and waste product removal or a spatial heterogeneity that perfusion strategies can avoid. Here, we implemented two protocols of frequency of medium delivery, $F = 1/24 \, h^{-1}$ and $F = 1 \, h^{-1}$ (Fig. 1f, top), that were simultaneously imaged in two independent microfluidic cell culture chambers (Fig. 1c). A flow rate of 3 µL/min was used to minimize the shear stress on the cell culture[47]. The temporal patterns of *Per2*::Luc mouse fibroblasts exhibited robust circadian expression for 6.5 days in both conditions, as shown in Fig. 1f. Interestingly, ultradian medium changes led to a shorter period of $22.6 \pm 0.8 \, h$, compared to

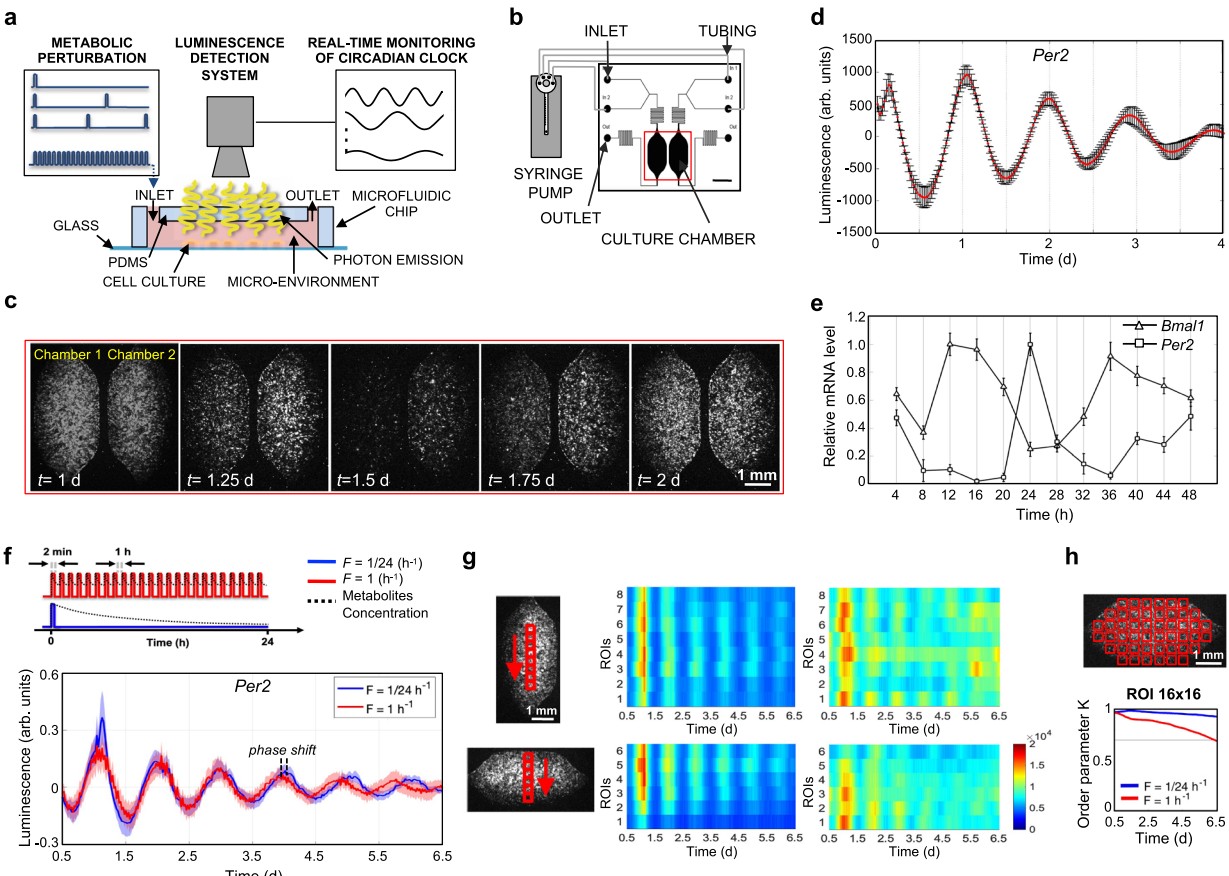

**Fig. 1 Real-time luminescence recording for live measurement of clock gene dynamics. a** Schematic representation of experimental setup for the acquisition of bioluminescence signal from cell integrated microfluidic chip; microfluidic setup is coupled with a microscope and liquid handling system. **b** Schematic representation of the liquid handling system connected to the microfluidic device, composed of two independent culture chambers (two inlets and one outlet for each). Scale bar 2.5 mm. **c** Time-lapse of bioluminescence images of a representative experiment, captured every 30 min, of mouse Per2::Luc fibroblasts, showing circadian rhythms of luminescence for 2 days. Scale bar 1 mm. **d** Baseline-subtracted Per2::Luc bioluminescence patterns acquired for 4 days from a microfluidic chamber after dexamethasone shock. Data are represented as mean ± s.d., $N = 9$ for each condition. **e** Temporal profiles of clock gene expressions, BMAL1 and PER2, in human fibroblasts integrated in the microfluidic platform, after synchronization ($t = 0$) with dexamethasone. mRNA level of clock genes was measured by qPCR and normalized to GADPH expression. Data are represented as mean ± s.d., $N = 3$ for each group. **f** Top, Schematic representation of the two different protocols of medium delivery imposed by automatic medium change. Bottom, Per2::Luc 6-day bioluminescence signal intensity in the 16-by-16 pixel ROIs, shown in the scheme in **h** (top), under the two medium change conditions, after baseline subtraction using LumiCycle Analysis program (Actimetrics). Solid line: mean signal intensity, patch: area delimited by mean ± s.d. of the 48 ROIs indicated in **h**. **g** Heatmaps of mean Per2::Luc 6-day bioluminescence signal intensity in the 16-by-16 pixel ROIs, shown in the schemes on the left, under the two medium change conditions in **f**. Colorbar indicates the bioluminescence signal intensity after image pre-processing in arbitrary units and applies to all four heatmaps. Scale bar 1 mm. **h** Kuramoto order parameter $K$, calculated for the time intervals between time 0.5 d and the time shown on the x axis, indicating the level of synchrony between the 16-by-16 pixel ROIs, shown in the scheme on top ($K = 1$ means complete synchrony). The horizontal gray line highlights a value of 0.7. Source data are available as a Source Data file. Scale bar 1 mm.

23.7 ± 0.3 h obtained with $F = 1/24\,h^{-1}$ (Supplementary Fig. S2a). We observed a consistent increase of the phase shift, measured as a time difference between two corresponding peaks in the two protocols ($F = 1/24\,h^{-1}$; $F = 1\,h^{-1}$), from 0.1 ± 1.8 h at day 1 to 6.3 ± 5.7 h at day 6 (Supplementary Fig. S2b).

To investigate the phase without any assumption of constant-period oscillatory behavior in either medium change regime, we analyzed the instantaneous phase using Hilbert transform[43] (Supplementary Fig. S2d, e). The parallel trends in both plots in Supplementary Fig. S2d demonstrate the constant-period hypothesis as reasonable, however, higher variability in the phase of different ROIs in $F = 1\,h^{-1}$ condition is showed in Supplementary Fig. S2e.

In order to ensure that circadian behavior was homogeneous throughout the cell culture surface and was independent from the specific region of the microfluidic cell culture chamber (typically, upstream/downstream or central/lateral position), we analyzed the images of the two culture chambers by discretizing the vertical and horizontal directions into adjacent ROIs of $16 \times 16$ pixels (Fig. 1g).

We found that the protocol with $F = 1/24\,h^{-1}$ ensured a more homogeneous distribution of the luminescence signal in horizontal and vertical directions, proving a well-defined circadian behavior. Conversely, the protocol with $F = 1\,h^{-1}$ was able to keep a good synchronization until day 2, after which the circadian gene expression became overall desynchronized.

In order to quantitatively analyze the synchronization over the whole microfluidic surface, we calculated the Kuramoto order parameter[42,44] for $16 \times 16$ pixel ROIs. The results in Fig. 1h confirmed that $F = 1/24\,h^{-1}$ provided a higher degree of spatial synchronization compared to $F = 1\,h^{-1}$. This result is independent from the ROI size as reported in Supplementary Fig. S2f, showing results for $8 \times 8$ and $32 \times 32$ pixel ROI.

In general, we can assume that the higher frequency of medium delivery ($F = 1\,h^{-1}$) led to a constant concentration of exogenous factors during an entire 24 h cycle (quasi-steady-state high-glucose condition). Whereas $F = 1/24\,h^{-1}$ induced dynamic changes of exogenous factors because of cellular uptake during 24 h cycle, which closely mimics typical oscillatory day/night behavior. Accordingly, we simulated this scenario with a simplified model of a representative metabolite concentration (Fig. 1f top).

**Exploring frequency and timing of feeding-fasting cycle.** The previous observations suggested that peripheral circadian clocks could be entrained by cyclic oscillations of metabolic cues and endocrine factors (e.g., glucose and insulin) associated with feeding and dietary regimens or food intake timing[48]. However, it is well known that the temporal trend of the concentration of these two molecules in the blood is highly correlated, as insulin acts in response to an increase of glucose concentration in the blood for maintaining energy homeostasis; thus, we reasoned that simultaneous variations of glucose and insulin concentrations could better mimic the physiological condition and be more relevant to study.

In order to enhance the capability of performing simultaneous experiments and increasing throughput, we integrated 14 microfluidic chips (Supplementary Fig. S8), each with 3 independent microfluidic cell culture chambers (Supplementary Fig. S1b), within a temperature-controlled luminescence imaging chamber equipped with a large field of view CCD camera. This experimental setup allows an integral measurement over the cell surface with a spatial resolution of 300 microns. Flow condition during media changes was either generated by volumetric pump using external liquid handling[49] or by capillary forces after media droplet deposition at microfluidic inlet[50].

With this setup, we asked how glucose and insulin, provided at different concentrations and frequencies, affect *Per2* expression in *Per2*::Luc mouse fibroblasts. First, we explored how different combinations of glucose-insulin affect *Per2* expression in *Per2*::Luc mouse fibroblasts in vitro (Supplementary Fig. S3a, b). This screen suggested that, while the period is unaffected (Supplementary Fig. S3c), a significant difference ($P < 0.01$) in the phase shift of *Per2* expression can be achieved by changing the concentration levels of glucose and insulin in the range of 2–25 mM and 0–50 nM, respectively (Supplementary Fig. S3d). Thus, to mimic the metabolic oscillatory behavior during day–night cycle, we will supply two types of media to the cells in the next experiments: medium H, containing 25-mM glucose (High concentration) and 10-nM insulin, and medium L, containing 2-mM glucose (Low concentration) and no insulin.

Next, we analyzed how the time at which the stimulus is provided affects the alignment of the internal biological clock with the external environment. We designed a series of experiments in which we switched between H (feeding) and L (fasting) media and vice versa within a 24 h cycle, modulating the exposure period of H and L media, in HL and LH respectively, as follows: 4:20 h, 8:16 h, 12 h:12 h, 16:8 h and 20:4 h (Fig. 2a).

After one cycle of feeding-fasting cycle, external environmental agents possibly affecting oscillations were absent, free-running (FR) stage, and self-sustaining rhythms were observed. Figure 2b shows *Per2*::Luc expression during metabolic perturbation and FR, for both HL and LH recorded from microfluidic chambers. Interestingly, the results show that the 12 h:12 h cycles between the two levels of glucose and insulin seem to provide the most sustained and robust circadian oscillations, compared to all other frequencies, in HL. On the other hand, LH, which is a misaligned metabolic pattern, exhibits disrupted oscillation of *Per2* during the stimulation part and higher variability in the FR observation (Fig. 2b). This experimental evidence is likely to be related to the alignment and misalignment between the time and the duration at which the metabolic cues are given and the intrinsic cell-autonomous clock. For all conditions, the stimulation was started at the trough of *Per2* (corresponding to $ZT = 6$, in the fasting phase in a murine peripheral oscillator model), which is aligned with HL stimulation and is in 12 h anti-phase with LH.

According to these results, we selected the 12-h frequency for all subsequent HL or LH cyclic experiments.

**Alignment and misalignment of metabolic cycles towards the core clock.** We asked whether the number of cycles of feeding-fasting influences the entrainment of the cell-autonomous clocks. We designed an experiment with an increasing number of cycles for both HL and LH stimulations (Fig. 3a, b, top). Figure 3a, b (bottom) show 6-day luminescence dynamics of *Per2*::Luc fibroblasts, stimulated with either HL or LH protocols for 1, 2, or 3 cycles. HL cyclic stimulations develop sustained and robust oscillations (Fig. 3a) with remarkable consistency between all conditions during FR. On the contrary, LH exhibits impaired oscillatory behavior and high variability during stimulation and regular oscillations restored during the FR (Fig. 3b) with a different phase distribution compared to HL (Fig. 3c). In addition, analysis of the instantaneous Hilbert phase (Supplementary Fig. S5a) showed that the period is constant between all conditions. On the second day of FR, we noticed that the variability of the instantaneous phase is higher in LH stimulations compared to HL, and this variability decreases in the condition of 3 cycles of stimulation.

We observed that *Per2* expression shows a phase advance in response to HL metabolic patterns whereas a phase delay to LH, with respect to the ZT before stimulation. Moreover, the increasing number of metabolic stimulation cycles increases the phase shift between HL and LH, reaching up to 8 h in the 3-cycle protocol (Fig. 3d).

We obtained comparable differences in terms of phase difference also in dish; we simulated metabolic oscillatory behaviors of day–night cycles by performing medium changes every 12 h for 3 days with the following media combinations: HL and LH, as in previous experiments, and also HH and LL, which mimic the ad libitum feeding and the low-calorie day–night regimes, respectively (Supplementary Fig. S4a). The recording of luminescence signal from the dish cell culture was possible only during the FR interval after the treatments. Supplementary Fig. S4b shows that *Per2* oscillatory behavior is significantly different across the four metabolic perturbations. Polar representation of the phase in Supplementary Fig. S4c shows that the largest difference in terms of phase shift was detected between HL and LH, the two anti-phase perturbations, with a phase difference of about 5 h maintained from days 5 to 7.

Since these results confirmed that 12 h:12 h HL stimulations of 24 h period can sustain the circadian *Per2* expression most effectively, we asked whether *Per2* oscillations are entrained by metabolic oscillatory patterns also with a period different from 24 h in Zeitgeber-dependent manner. To answer this question, we designed a specific experiment in which we changed the period of feeding and fasting cycles ($n = 3$) from 24 to 20 h and 26 h, with HL stimulation aligned with the circadian phase ($ZT = 6$). As controls, we used HH stimulations, where the fasting is not present, and dexamethasone shock, as a reference for circadian clock synchronization. Figure 3e shows that HL cycles of 24 h period, together with HH stimulations (ranging from 20 to 26 h period) and dexamethasone shock, resulted in sustained *Per2* oscillations during FR. *Per2* oscillations for HL conditions of

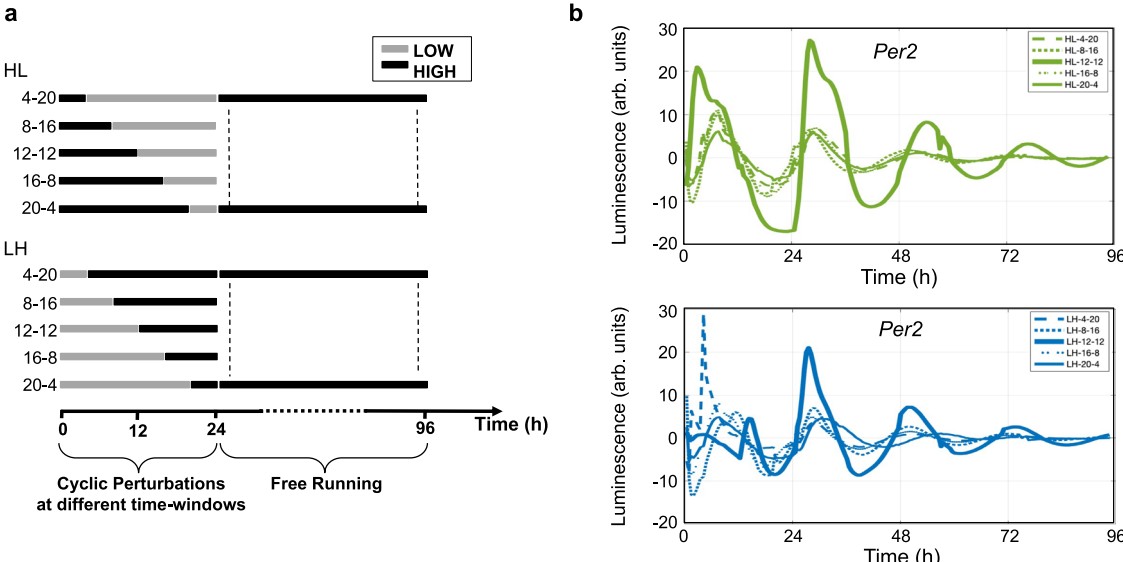

**Fig. 2 Circadian behavior under different time-dependent cyclic perturbations. a** Schematic representation of daily perturbations under HL or LH cycles imposed at different time-windows (4:20 h, 8:16 h, 12:12 h, 16:8 h, and 20:4 h), followed by a period of FR. **b** Baseline-subtracted Per2::Luc bioluminescence rhythms of HL and LH imposed at different time windows, was acquired from $t = 0$ to $t = 96$ h at intervals of 30 min; data are represented as mean of bioluminescence signal after baseline subtraction, $N = 3$ for each group. Source data are available as a Source Data file.

20 and 26 h period were damped after 2 days of FR. The period analysis of *Per2* oscillations shows that dexamethasone shock, together with all HH conditions and 24 h HL stimulation, maintain a *Per2* oscillation period of 24 h (Fig. 3f). Consistently with previous results, the absence of fasting phase in the cyclic stimulation (HH), even with period of 20 and 26 h, results in maintenance of the 24 h intrinsic *Per2* rhythm. On the other hand, the entrainment with feeding and fasting cycles of 20 and 26 h periods, showed in both cases *Per2* oscillation period associated with high variability and with signal damping as shown in Fig. 3f and in Supplementary Fig. S5b reporting the instantaneous phase.

Thus, in the protocols with the fasting phase, the synchronization between the Zeitgeber metabolic entrainment and the peripheral clock is sustained only when HL is aligned with the intrinsic period of the peripheral circadian clock of the cells (24 h). It is possible that, in presence of the fasting phase, HL stimulation, initially aligned with Per2 expression, progressively diverges toward misaligned conditions, after few cycles, in the case of 20 and 26 h period, and are likely outside the range of entrainment for these cells.

**Transcriptomic landscape of cyclic metabolic patterns**. Thus, we asked whether the synchronization between the feeding-fasting cycles and the cell-autonomous clock (phase of *Per2*) affects the entire transcriptome. We performed the experiment described in Fig. 4a. Briefly, after 3 days of cyclic treatments of HL, LH, and HH, and 12 h of FR, RNA samples were collected every 4 for 24 h, and transcriptomic analysis was performed. Figure 4b summarizes the number of transcripts that display a circadian rhythmicity, selected using the nonparametric algorithm JTK_cycle[51] with a *P*-value < 0.01 in each condition. A similar number of genes with oscillatory behavior were identified: 304 genes in HL, 313 in LH, and 278 in HH. Surprisingly, the overlap of oscillatory genes between conditions was low: ~16% of genes common to pair of conditions, and only 9% shared among all three conditions (Fig. 4c), suggesting that the three metabolic protocols are able to entrain the gene expression in different ways. The 27 genes at the intersection include components of the main

clock (*Cry1, Bmal1, Rev-Erbα/β*), genes related to extracellular matrix (*Col4a2, Col6a2, Fbln2, Lox, Mgp, Spon2*), and to cell division (*Aspm, Kif2c, Lig1, Ncapd2, Nusap1, Smc2, Top2a, Wee1*). Thus, despite the three protocols are all able to induce a circadian entrainment, there could be relevant functional differences among them. We found very few genes that significantly changed their mean temporal expression among conditions, and, of these, only two (*Ptcd2, Dner*) had an oscillatory behavior only in LH condition (Fig. 4d). Common oscillating genes have similar amplitude but show some phase differences (Fig. 4e) associated with an enrichment of different functionalities (Supplementary Fig. S6a). This result suggests that the metabolic cycles imposed have an impact on the phase rather than the amplitude. Heatmaps, shown in Fig. 4f, display circadian genes found in HL (left panel), LH (middle panel), and HH (right panel) and also the gene comparison among the experimental conditions (Supplementary Fig. S6b). Another general feature of the oscillating transcriptome is that the LH condition tends to synchronize all genes with the same phase (~4 h), while HL and HH entrain gene oscillations with a higher variety of phases (Fig. 4f and Supplementary Fig. S6c). Some genes (*Cry, Bmal1, Rev-Erbalpha, Bhlhe41*) of the core circadian clock show robust oscillatory behavior with respect to metabolic perturbations, suggesting the independence of their oscillatory behavior from metabolic cues (Fig. 4g). In particular, considering eight core-clock genes, seven of these oscillate in the HL condition with *P*-value < 0.01, whereas only five and six in LH and HH conditions, respectively. Analysis of KEGG pathways enriched in the gene subsets oscillating in each condition (Fig. 4h), shows that circadian rhythms is the only category common to all, although more significant in HL and HH. HL shows enrichment of cell cycle, whereas LH seems to regulate by circadian genes ECM production and protein digestion and absorption (also shown in Supplementary Fig. S6d), which are the major functions of fibroblasts.

Polar diagrams highlight the peak time distribution of circadian genes in the 3 conditions (Fig. 5a). Most of the circadian genes display a peak in the light phase [0–12]: HL 76% [0–12] vs. 24% [12–24], LH 87% [0–12] vs. 13% [12–24] and HH 63% [0–12] vs. 37% [12–24]. Interestingly, HL and LH show a

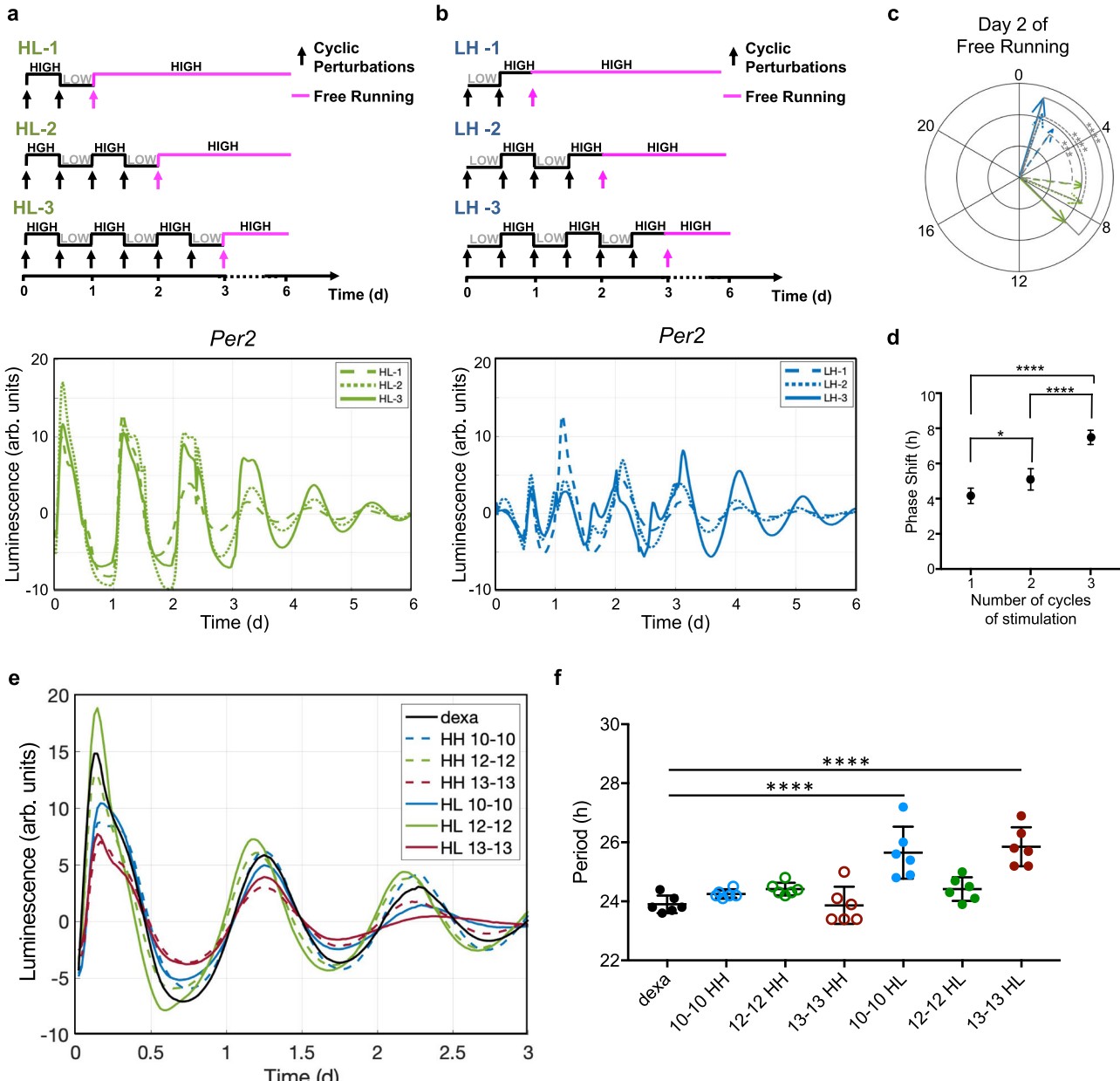

**Fig. 3 Metabolic behavior of cyclic day and night by microfluidic technology. a** Schematic representation of feeding-fasting (HL) behavior imposed for 1, 2, or 3 days and followed by a FR condition (top); baseline-subtracted Per2::Luc bioluminescence patterns of Per2::Luc mouse fibroblasts in HL-1,2,3, acquired at an interval of 30 min (bottom). Data are represented as mean, $N = 3$ for each group. **b** Schematic representation of fasting-feeding (LH) behavior imposed for 1, 2, or 3 days and followed by a FR condition (top); baseline-subtracted Per2::Luc bioluminescence patterns of Per2::Luc mouse fibroblasts in LH-1,2,3, acquired at an interval of 30 min (bottom). Data are represented as mean, $N = 3$ for each group. **c** Polar representation of the relative temporal position of the acrophases of HL-1,2,3 and LH-1,2,3 profiles at day 2 of FR, $N = 6$ for each group; the arrow length (radial axis) indicates the value of amplitude, the angular position indicates the position of acrophase. The standard deviations of the acrophases in each condition are 0.52 h (HL-1), 1.04 h (HL-2), 0.52 h (HL-3), 1.97 h (LH-1), 1.57 h (LH-2) 0.52 h (LH-3); ***$P = 0.0005$, ****$P < 0.0001$ two-sided $t$-test. **d** Comparison of the phase shift between the HL and LH conditions imposed for 1, 2, or 3 days, after 48 h of FR. Data are represented as mean ± s.d., $N = 6$ for each group. *$P = 0.0124$, ****$P < 0.0001$ one-way ANOVA with Tukey's multiple comparisons test. **e** Baseline-subtracted Per2::Luc bioluminescence rhythms during FR after 3 cycles of stimulation of HL and HH imposed with different time windows (10:10 h, 12:12 h, 13:13 h). The luminescence was acquired from $t = 0$ to $t = 3$ d at intervals of 30 min; data shown are mean baseline-subtracted luminescence signal, $N = 4$–6 for each condition. **f** Period comparison among the experimental conditions implemented in **e**, ****$P < 0.0001$, using one-way ANOVA with Tukey's multiple comparisons test. Source data are available as a Source Data file.

higher abundance of genes with a peak at 4 or 6 h with a percentage of 35.5% and 72.5%, respectively, compared to the total number of genes. HH displays a lower polarization, with most genes clustering with a peak at 8 or 12 h with a percentage of 31.3% compared to the total number of genes.

Finally, we functionally characterized the genes with oscillatory behavior in the three metabolic entrainment conditions. As mentioned above, the majority of entrained genes in the three protocols is different. Here, we also show that they are enriched in different categories with strong differences between genes peaking

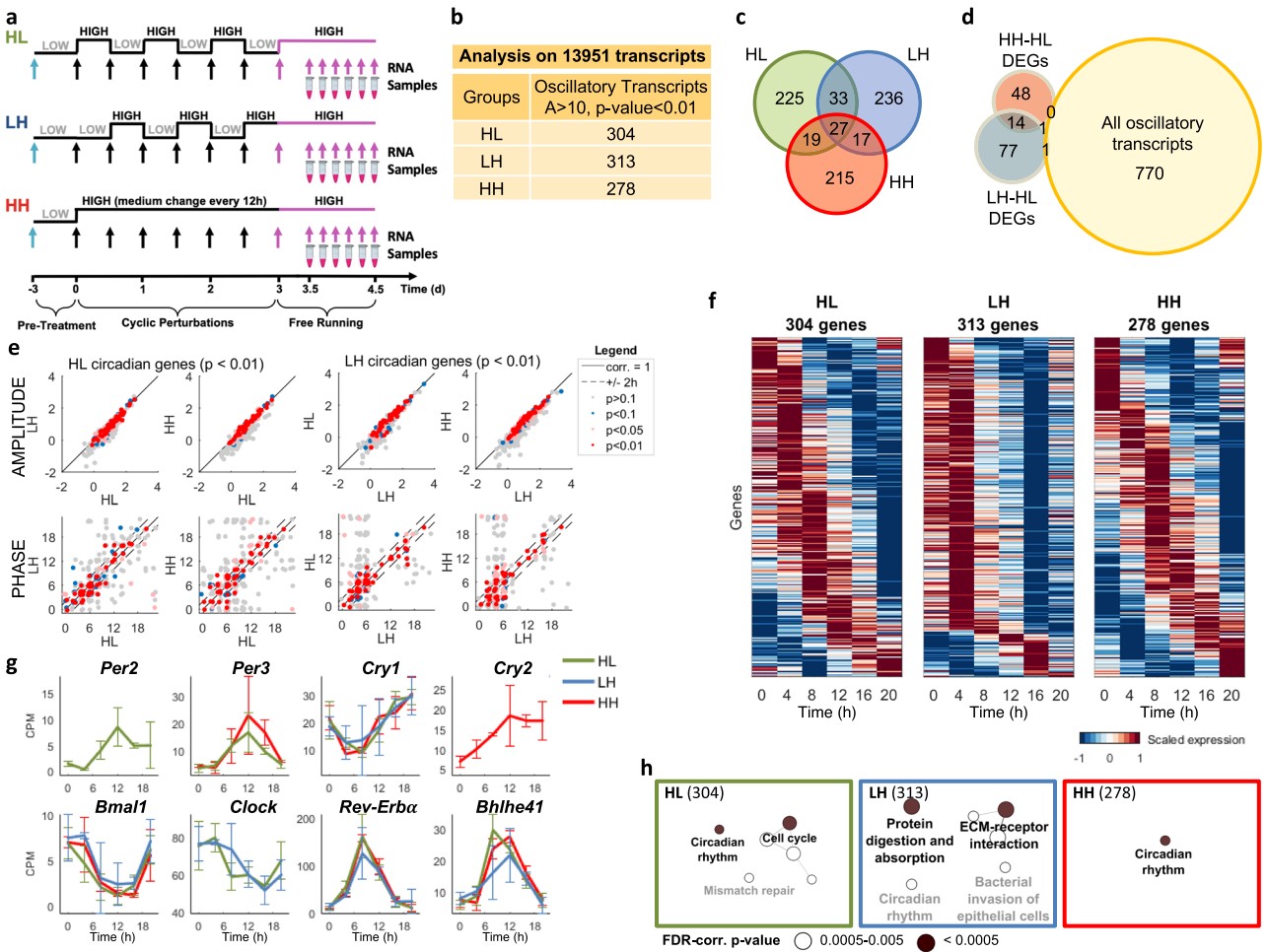

**Fig. 4 The circadian transcriptome is reprogrammed by cyclic metabolic patterns. a** Schematic representation of feeding-fasting (HL), fasting-feeding (LH), and ad libitum feeding-like (HH) conditions imposed for 3 days and followed by a FR condition. **b** Number of circadian genes by JTK analysis, with Amplitude > 10 and *P*-value < 0.01, both provided by JTK test. **c** Venn diagram including circadian genes (*P* < 0.01 of JTK test) in each condition. **d** Venn diagram including all circadian genes displayed in **c**, and HH-HL and LH-HL DEGs. **e** Correlation of circadian gene amplitude and phase, obtained by JTK algorithm, between pairs of conditions. The indicated *P*-values refer to the significance level of circadianity. **f** Heatmaps of oscillating transcripts (*P* < 0.01 of JTK test) under HL, LH, and HH conditions. Genes are ordered by phase. **g** Expression of core clock genes in the conditions indicated in the legend. Only genes with circadian behavior (*P* < 0.01 of JTK test) are shown. Error bar: mean ± st. dev, N = 4 for each condition. **h** Enrichment analysis results of genes oscillating in HL, LH, and HH conditions within pathways in the KEGG database. Links between categories represent the κ-value.

during the active or resting phases (Fig. 5b). The oscillating transcripts having a peak during the active phase, all show a significant enrichment in the KEGG pathway "circadian rhythm". On the contrary, transcripts peaking during the resting phase show major functional differences and are enriched in different metabolic and also non-metabolic pathways. HL condition especially entrains genes related to cell cycle and senescence pathways, while LH condition is related to cell–ECM interaction. Among the three most enriched categories, shown in Fig. 5c, genes related to extracellular matrix (*Col4a1, Col5a3, Col6a1, Col4a2*) are characterized by a highly significant oscillatory profile with a −log(*P*-value) up to 5.5 with phase peak aligned with the resting phase.

## Discussion

Our study provides a comprehensive and detailed analysis of the effects of different schedules of feeding-fasting cycle on the in vitro peripheral clock entrainment, which was previously studied especially using in vivo models[52].

The microfluidic setup specifically designed to study the circadian clock enables precise control of the cellular environment

with accurate flow rates while maintaining the possibilities of using reporter cell lines and transcriptional analysis including sequencing data. Despite the increasing evidence of potential for dynamically perturbing cell culture accurately[53], only a few studies have reported the use of microfluidics for studying the circadian clock, mainly limited to algae and fungi, without taking into account oscillatory perturbations[39–41].

We thus developed a microfluidic system for performing a variety of dynamics of metabolic perturbations under otherwise unperturbed conditions. The dynamical control of the in vitro environment allowed us to investigate the role of the frequency and the time at which the metabolic cues are provided. In particular, the media change with high frequency leads to an impairment of the clock whereas a more coherent frequency with the 24 h day and night cycle maintains the cell-autonomous clock robustness. Our data show that feeding and fasting must be imposed in a particular window of time and also with a specific frequency, to keep a sustained circadian oscillation, confirming the importance of the Time-Restricted Feeding to improve health and longevity[6,25,27].

Surprisingly not only the timing of feeding-fasting is important, as shown with the 12 h:12 h pattern, but also the alignment

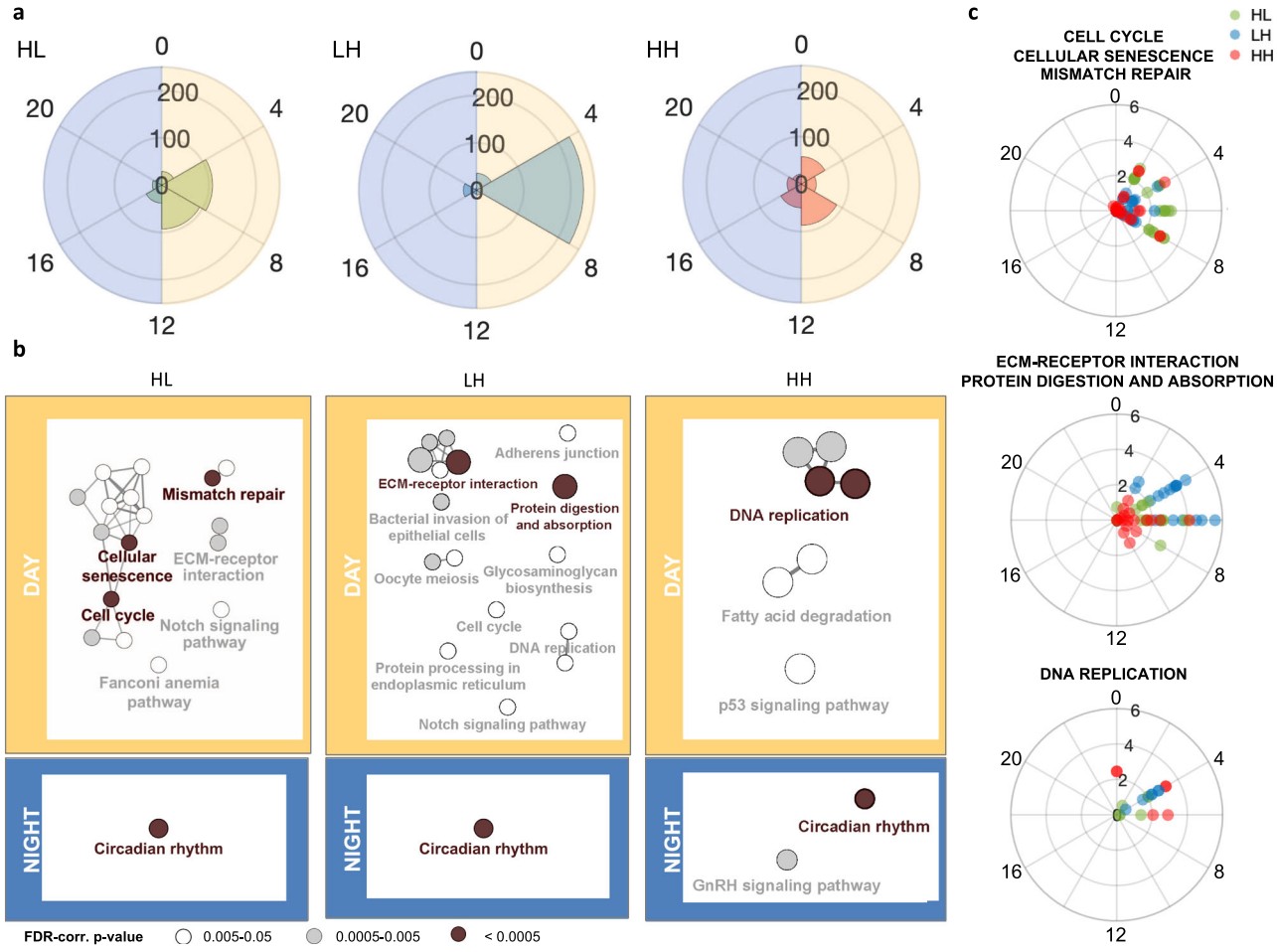

**Fig. 5 Day and night entrainment of the transcriptome. a** Polar diagrams of the phase distribution in HL, LH, and HH. Light = 0–12 h, Night = 12–24 h. **b** Enrichment analysis of genes oscillating in HL, LH, or HH, having their peak during the 12-h day (top) or the 12-h night (bottom), within pathways in KEGG database. **c** Phase and −log10(P-value) of genes belonging to the most enriched KEGG pathways during the resting phase in HL, LH, and HH, respectively.

between metabolic signals and circadian rhythms is crucial, to guarantee a sustained oscillation of the *Per2* gene. Moreover, also the interval between feeding and fasting phases can have a crucial effect in maintaining the circadian oscillation self-sustained: when the time of these two phases is reduced or increased, *Per2* rhythm undergoes an amplitude damping and the period raises. It is well-known from previous observations that the timing of food intake is a powerful synchronizer of many peripheral clocks, such as the liver[5,6,20], and can influence body weight and other health parameters[54].

Indeed, in our daily routine, we are continually exposed to feeding and fasting cycles, and it is important to take into consideration the overall coherence between metabolism and circadian rhythms[12,14]. The loss of such a coherence, translating in a misalignment, may contribute to the weakening or desynchronization of oscillators.

These results have been confirmed when multiple cycles of feeding and fasting have been provided. The correlation between metabolic fluctuations and circadian oscillators was also evaluated for different temporal durations (1, 2, and 3 days), leading to an increase of phase difference of *Per2* peaks between the two opposite metabolic patterns HL and LH.

On the other hand, in presence of both feeding and fasting phases, the synchronization between the period of Zeitgebers metabolic entrainment and the peripheral clock is sustained only when feeding and fasting schedules are continuously aligned with the intrinsic period of the cells.

These results are reminiscent of Pittendrigh's resonance hypothesis in which driving cycles that are close in the period to the circadian oscillator enhance the amplitude of the circadian oscillation[55].

All these findings confirm that regulation of metabolism by the circadian clock and its components is reciprocal, attesting that metabolism and circadian clocks are tightly interlocked: clocks drive metabolic processes, and various metabolic parameters affect clocks, producing complex feedback relationships[56,57]. Recent studies have shown that a high-fat diet in mice leads to changes in the period of the locomotor activity rhythm and alterations in the expression and cycling of canonical circadian clock genes, nuclear receptors that regulate clock transcription factors, and clock-controlled genes involved in fuel utilization in peripheral tissues[22]. Furthermore, gene expression and metabolomics profiling have revealed that a defined daily period of feeding and fasting is a dominant determinant of diurnal rhythms in metabolic pathways[17,58,59]. The duration of feeding and fasting, and consequently the period of the cycles, is likely to determine the overall anabolic (feeding) and catabolic (fasting) signals needed to maintain the body in healthy condition. Consistently with this hypothesis, Chaix et al. varied the duration of food access to characterize the temporal boundaries within which time-restricted feeding benefits persist. They demonstrated that time-restricted feeding induces a therapeutic effect in terms of improving glucose tolerance, nutrient homeostasis, and reducing insulin resistance proportionally to the duration of fasting[60].

Thus, it would be interesting to evaluate the impact that different feeding-fasting regimes have on the entire transcriptome. Other colleagues showed only a few cycling transcripts ($N = 11$) present in NIH3T3 and U2OS cells after synchronization with either forskolin or dexamethasone, respectively[61]. Remarkably, in the metabolic entrainment proposed in this work and based on sustained 3 days of feeding and fasting cycles, RNA-sequencing data from this work show a significant number of oscillating genes ($N = 500$) belong to the transcriptome in all the dietary regimes. This result could be the consequence of the cyclical nature of the metabolic perturbations implemented in our study. Importantly, oscillating genes do not show differential expression in terms of amplitude but in phase, highlighting the role of the alignment and misalignment between metabolic habits and clock regulation.

The circadian rhythms category is more significant in HL and to less extent in LH, suggesting that feeding-fasting cycles mimicking the physiological conditions are optimal for maintaining the oscillatory nature of circadian genetic circuits. Our investigation has also shown that the fasting phase is important to have a phase polarization of the transcriptome, which is completely missing in the HH condition where the cells were always kept under a high level of nutrients. It is worth mentioning that the misalignment leads to a polarized expression of ECM transcripts, which is a typical mechanism of activated fibroblasts that is frequently associated with fibrosis[62]. It was observed that this polarization involves different pools of genes and as consequence is enriched for different metabolic functions.

In conclusion, the data presented here identify microfluidics as a promising and crucial technology for future studies of dynamic processes offering a new perspective of study in the circadian field. Our results corroborate several observations of the implication between metabolism and clock[63] and provide additional evidence that alignment and misalignment between cell-autonomous clock and the feeding-fasting cycles can provide not only alteration on the clock but also in other functions.

A corollary of these findings is that not only do conditions associated with an alternating cycle of feeding and fasting result in transcriptional polarization. The alignment and misalignment between clock and external cues, in turn, leads to transcriptional changes, strongly suggesting that timing and phase play an important role in regulating the cellular function and more in general the tissue homeostasis. Understanding the integration of aligned and misaligned metabolic habits with the circadian clock may ultimately contribute in preventing the onset of associated severe diseases as already demonstrated for diabetes and cancer[12,14] and infected diseases[64,65].

## Methods

**Microfluidic devices.** In this work, we developed two microfluidic devices, both fabricated by soft lithography technique and replica molding, previously published by our group[66]. Polydimethylsiloxane (PDMS) with a 10:1 base/curing agent ratio (Dow Corning) was coupled to a borosilicate glass slide (Menzel–Gläser) through plasma treatment of surfaces.

The first microfluidic platform, used for luminescence signal acquisition with high spatial resolution, consists of two independent culture chambers, with the following dimensions: 7.0 mm of length, 2.5 mm of width, and 0.1 mm height with a 1.75 µL volume (Supplementary Fig. S1a). With a ×10 objectives, and camera resolution of 256 × 256 pixel, in a field of view of 5.6 × 5.6 mm one, for a final resolution of ~20 microns. Each culture chamber is connected to two independent inlets and one outlet by a 26.5 mm long serpentine, 2.5 mm wide and 1.7 mm high, that is able to avoid diffusive transport during the cyclic metabolic perturbations. Inlets and outlets are connected, respectively, to a multi-port syringe pump (XLP pump model, Tecan, Switzerland) and waste reservoirs by Teflon tubes (0.3 mm ID, 0.8 mm OD, Cole Parmer). The remote controller of the multi-port syringe pump allows either continuous or discontinuous perfusion of the microfluidic cell culture with two different media. Pumps were controlled through a serial port by customized software developed in LabVIEW (v.2019, National Instruments) in order to set flow rate, duration of perfusion, and pause and cycles number.

Our LABVIEW code is publicly available (https://zenodo.org/badge/latestdoi/337143716).

The second device was designed to acquire a high-content luminescence signal. Each microfluidic chip includes 3 parallel culture chambers with a volume of 2.7 µL each, and one inlet and one outlet (Supplementary Fig. S1b). The microfluidic chamber design has the following dimensions: 18.0 mm of length, 1.5 mm of width, and 0.1 mm height. The field of acquisition 15 × 15 cm can allow to host 14 chips of 3 independent channels. As in the first device, inlets are connected to a multi-port syringe pump (XLP pump model, Tecan, Switzerland) for automatic medium delivery of two different media in a frequency manner. Alternatively, a different diameter of the microfluidic inlet and outlet port (respectively, 1 and 3 mm), guarantee a capillary flow for fast media changes (0.2 µL/s) by deposition of 4 µL media droplet over inlet port and simultaneous empty of outlet port, as detailed explained in our previous work[50].

Both types of devices are sterilized by autoclaving before use. During experiments the microfluidic chips are placed in a dish, surrounded by a water bath to reduce medium evaporation.

**Cell culture and media.** For the experiments, two cell lines are used: HFF (ATCC, Cat. SCRC-1041) and *Per2::Luc*, immortalized mouse-ear fibroblast cell line carrying a PER2::LUCsv bioluminescence reporter generated from *Per2::lucSV* knockin mice[67,68]. The cells were expanded in Dulbecco's Modified Eagle's Medium, with high-glucose content (4.5 g/L) (Life Technologies), 10% FBS (Life Technologies), and 1% penicillin/streptomycin (Life Technologies) at 37 °C and with humidified 5% $CO_2$. Passaging of both cultures was performed with 0.25% trypsin-EDTA (Life Technologies) and cells were either re-plated on culture dishes for further expansion or seeded inside microfluidic chambers.

Microfluidic cell cultures were performed as follows. The cells were seeded in the microfluidic chambers, at a density of 600 cell/mm$^2$, after a coating with 25 µg/mL of cold fibronectin (Sigma Aldrich); this density allowed the confluence achievement the day after seeding. Before placing in incubator, 1 ml of PBS 1× was added to the bottom of the dish, in order to maintain proper humidity.

The low glucose-low insulin (L) medium was prepared with 2 mM glucose, while the high glucose-high insulin (H) medium contains 25 mM glucose and 100 nM insulin (Sigma Aldrich).

**Circadian synchronization on-chip.** Three different protocols were used for synchronizing the cellular model integrated into the microfluidic platform.

One protocol is based on a dexamethasone shock. At time $t = 0$ h, the cell culture was treated with a medium containing 100 nM dexamethasone, which was replaced with serum-free DMEM after 2 h. In the second protocol, a serum shock (i.e., medium containing 10% FBS) at time $t = 0$ h was used for synchronizing the cells. At the different time points, the cells were lysed and mRNA samples were extracted from the microfluidic channels for performing quantitative RT-PCR, as described below.

The third one is based on the use of H and L media changed at different frequencies. The media contain Dulbecco's Modified Eagle's Medium (Corning) with different content of glucose and insulin, 1% penicillin/streptomycin (Life Technologies), 10 mM of HEPES (Life Technologies), and 350 mg/L of sodium bicarbonate (Sigma Aldrich) as a buffer, and 1 mM of luciferin (Biosynth). All media were supplied with 1% FBS and 2% B27 Supplement minus insulin (Life Technology).

**Quantitative RT-PCR.** The lysis buffer RT-qPCR iScript™ Sample Preparation Reagent (BIO-RAD) was used for extracting mRNA samples from the microfluidic channels every 4 h. Before extraction, cells must be washed gently with cold PBS 1× twice. 10 µL of iScript was injected into a microfluidic channel and left for 1 min at room temperature. All reagent was collected in a 0.2 mL RNase-free tube kept on ice. Samples were stored at −80 °C.

mRNA was reverse transcribed with the High Capacity cDNA Reverse Transcription Kit (Applied Biosystems), with its recommended protocol.

Real-time PCR was performed on the ABI PRISM 7000 Sequence Detection System (Applied Biosystems) using TaqMan Gene Expression Master Mix (Applied Biosystems) and pre-designed TaqMan primers according to the recommended protocol. Fold changes in expression of *PER2* (HS 00256143_m1, ThermoFisher Scientific) and *BMAL1* (HS 00154147_m1, ThermoFisher Scientific) for each sample, normalized to *GAPDH* (Hs02758991_G1, ThermoFisher Scientific) expression, were determined using the $2^{-\Delta\Delta CT}$ method. The complete list of all primers used, including the names and sequences, is reported in Supplementary Table 1.

**Luminescence data acquisition.** Luminescence images were acquired by both CCD camera, Series 600s or 850s (Spectral Instrument), and photomultiplier tubes, depending on the experimental setup.

The 600s CCD camera, installed at the bottom part of an inverted microscope, measured the photon emission from the cell culture with high spatial resolution over 5.6 mm × 5.6 mm of acquisition field; images with 30 min exposure time, were continuously acquired up to 6 days. The 850 s CCD camera was mounted on the top of Xenogen IVIS Imaging System 100 Series (Perkin Elmer).

Bioluminescence images with exposure time of 30 min over 15 cm × 15 cm area were continuously collected using Living Images Software (v. 4.3.1.0.16427, Perkin Elmer). All acquisitions were performed in complete dark conditions. A custom-engineered heated chamber was developed to fit around the microscope stage and inside the IVIS, keeping the cells at constant 37 °C; sodium bicarbonate and HEPES served as buffers[34].

Bioluminescence data from cell culture on 35-mm culture dishes were collected by LumiCycle (Actimetrics Inc., Evanston, IL) with a 32-channel carousel unit. This system is equipped with 4 photon-counting photomultiplier tubes, each selected for low dark counts and high sensitivity in the green portion of the spectrum at which luciferase emits light. Because it has no thermo-controller and is placed in an incubator and handled by a computer. The bioluminescence signal from each dish was measured for 70 s at intervals of 10 min.

**Imaging processing**. The imaging processing requires the following steps: correction from artifacts, analysis, and quantification of the circadian parameters.

First, we performed image pre-processing using ImageJ v. 1.53c. Images were imported and arranged within an image stack to apply the same corrections to every image. Cosmic ray artifacts were removed using a selective median filter (function *Process > Noise > Remove Outliers* with default parameters in ImageJ). Then histogram equalization was performed (function *Image > Adjust > Brightness/Contrast* with default parameters in ImageJ), as already reported in our previous publications[67,69]. Finally, images were exported as 16-bit.tif files.

Further image analysis and result visualization were performed using MATLAB R2020b. Non-overlapping 16-by-16 pixel ROIs were manually selected within the surface of the microfluidic chamber as shown in Fig. 1g. Other ROI sizes (8-by-8 and 32-by-32 pixel) were also analyzed. 16-by-16 pixel ROIs represented a trade-off between having a high number of ROIs per chamber and include a number of cells that are representative of bulk population response. This trade-off was evaluated by calculating the Kuramoto order parameter[42,44], which shows the synchrony of the oscillatory behavior between different ROIs. Mean ROI bioluminescence intensity was calculated after the image pre-processing described above for each time point to get a temporal profile.

The baseline subtraction was performed with two different methods. The first, used for all the figures, is based on LumiCycle software (v. 3.002, Actimetrics Inc., Evanston, IL) to subtract the baseline that was calculated as a running average of the data; the baseline curve is subtracted from the raw data to derive baseline-subtracted curves[69].

The second, used to verify that the previous one did not introduce artifacts in the main oscillatory behavior, was used only in Supplementary Fig. S2c, and subtracts a baseline obtained by an average of the data within a centered moving window corresponding to 24-h intervals (the approximate expected period of data oscillations)[70].

Hilbert transform was calculated on the baseline-subtracted signal to estimate the instantaneous phase of the oscillatory profiles. Moreover, the continuous Hilbert phase was calculated according to Caranica et al. [43], which was then used for calculating the Kuramoto order parameter, $K$[71,72].

For the heat map in Fig. 1g, a script was developed using MATLAB2020b, to visualize spatial heterogeneity both in the longitudinal and transversal directions of the chambers. 8 and 6 non-overlapping 16 × 16 ROIs were defined in the chamber in the longitudinal and transversal directions of each chamber, respectively (as shown in Fig. 1g). The average intensity was calculated and plotted for each ROI using images pre-processed as above described. Our Matlab code is publicly available (https://zenodo.org/badge/latestdoi/337143716) and freely available for data processing.

The period in Supplementary Fig. S2b and Fig. 3g was calculated as the dominant period in the baseline-subtracted data fitted with the best-fit damped sine wave multiplied by the best-fit negative exponential, using LumiCycle Analysis software (Actimetrics Inc., Evanston, IL) by the sin fit damped method.

The phase shift was obtained as the peak-to-peak time difference of two corresponding peaks in two different experimental conditions.

**RNA-seq and bioinformatic analysis**. Bulk RNA-seq data presented in this study have been deposited at the Gene Expression Omnibus database (https://www.ncbi.nlm.nih.gov/geo/query/acc.cgi?acc=GSE167763) with the data set identifier GSE167763.

Total RNA was extracted using RNeasy Plus Micro Kit (Qiagen), following manufacturer-suggested protocol. Total RNA was quantified using the Qubit 2.0 fluorimetric Assay (ThermoFisher Scientific). Libraries were prepared from 100 ng of total RNA using a 3′DGE mRNA-seq research-grade sequencing service (Next Generation Diagnostics srl)[73], which included library preparation, quality assessment, and sequencing on a NovaSeq 6000 sequencing system using a single-end, 100 cycle strategy (Illumina Inc.). The raw data were analyzed by Next Generation Diagnostics srl proprietary 3′DGE mRNA-seq pipeline (v2.0), which involves a cleaning step by quality filtering and trimming, alignment to the reference genome, and counting by gene[74,75].

The library has been processed two times and samples with less than 2500 genes above 5 cpm were excluded from the analysis. The technical replicates were pulled together. After that, we have filtered out all genes having < 1 cpm in less than three samples. Differential expression analysis was performed using edgeR[76].

DEG genes between HH-HL and LH-HL conditions were selected by ANOVA with $P < 0.01$ followed by multiple comparison test with $P < 0.05$, considering the mean expression profile.

CPM data were scaled in the interval $[−1, 1]$ for visualization in heatmaps, and genes were ordered according to their phase obtained from JTK[51].

Over-representation analysis within KEGG pathway or Gene Ontology databases was performed using ClueGO v. 2.5.4[77], and visualized within Cytoscape v. 3.5.1[78] or as a bar plot in MATLAB. Other plots were also performed in MATLAB.

**Statistical analysis**. For each experimental condition, at least three independent biological experiments were performed. Statistical significance was assessed by the two-sided $t$-test and one-way ANOVA with Tukey's multiple comparisons test, and presented as the mean ± s.d, determined from at least three independent experiments. For all analysis: *: $P < 0.05$; **: $P < 0.01$; ***: $P < 0.001$; ****: $P < 0.0001$. All $N$ values defined in the legends refer to biological replicates unless otherwise indicated. GraphPad Prism Mac (v. 7.0a) was used to do graphs and statistical analysis.

**Reporting summary**. Further information on research design is available in the Nature Research Reporting Summary linked to this article.

## Data availability

The authors declare that all data supporting the findings of this study are available within the article, its Supplementary Information, attached files, and online deposited data. RNA-seq data can be accessed at "GSE167763"[79], or from the authors upon reasonable request. Source data are provided with this paper.

## Code availability

All the codes developed in this works are deposited in GitHub repository and can be enduringly and freely accessed. The LABVIEW code is publicly available (https://zenodo.org/badge/latestdoi/337143716[80]). The Matlab codes are publicly available (https://zenodo.org/badge/latestdoi/337143716[81]) and freely available for data processing.

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

## Acknowledgements

This research was supported by Progetti di Eccellenza Ca.Ri.Pa.Ro. and Progetto Wild Card of University of Padova. J.S.T. is an Investigator in the Howard Hughes Medical Institute. O.G. was supported by Fondazione Umberto Veronesi. C.L. was supported by grant F-0301-15-009 by ShanghaiTech University.

## Author contributions

O.G., J.S.T., and N.E. designed the study. O.G. performed all the experiments and data analysis. Y.L. helped with the high-resolution luminescence imaging. S.A. and W.Q. helped in microfluidic experiments. F.P and D.C. performed RNA-sequencing. C.L. performed Hilbert transform and bioinformatic analysis. O.G., J.S.T., and N.E. critically discussed the data and wrote the manuscript. J.S.T. and N.E. supervised the project.

## Competing interests

The authors declare no competing interests.
