## [Peer Review File · Nature Communications]

Reviewers' Comments:

Reviewer #1:

Remarks to the Author:

Summary: The microfluidics approach is very interesting; however, the work is not tied into the clock literature, either experimental or theoretical. The two parts of the paper are not logically connected because one part is on the microscopic scale of single cells, and the other part is on the macroscopic scale through standard RNA transcriptomics. The authors do not make use of the continuous time measurement at the single cell level. The description of the microfluidics needs to be more complete in Materials and Methods. They do not characterize the level of detection noise in their single cell measurements, leaving in question the validity of results presented in the microfluidics experiments. The phase analysis is incomplete and not contemporary.

The connection to existing models of the clock is absent. For example, if the cells constitute damped oscillators, then one might argue they can entrain to any period of the driving input. On the other hand if single cells have their own intrinsic clock, they can entrain only within a range of frequencies about the intrinsic frequency of each cell. From the data presented it is not clear that single peripheral cells have a clock. Not knowing that individual cells have clocks leaves open how to explain the entrainment phenomenon observed.

Line number

43. What is missing in the introduction is a relation of the work to other model systems on the clock, such as in Arabidopsis, where glucose entrainment was reported earlier in Nature. What is also missing is the first global descriptions of the clock network in other model systems to help interpret the transcriptomics data.

80. The microfluidic approach is to be commended, but their microfluidic approach to the entrainment problem is not novel and has been utilized in other model clock systems, but the authors cite none of this work.

104. This measurement of clock gene luminescence in single cells in an entrainment experiment is not the first time this has been done. This has been done in other clock systems, but again it would be useful to relate the results here to work on other clock systems. For example, how did the authors go about separating the detection and stochastic intracellular noise in the luminescent measurements made?

105 What fraction of the variation in luminescence is detection noise? How is the reproducibility of the measurements quantified? What controls have been done with luminescent beads substituted for living cells and the like? Without the characterization of the detection noise the microfluidic results remain in question.

107 The authors have not taken advantage of this system and shown continuous data acquisition. Why have they not done so, if the systems is so reproducible? How do the findings on Per2::Luc compare with homologs in other systems, where single cell measurements have been taken on the clock mechanism?

108. They authors assert high resolution at single cell level. How were cells identified?

127. The authors are not using reasonable and well studied measures of phase, such as the Hilbert phase. These contemporary phase measures like Hilbert phase allow one to get away from assuming constant phase with the phase shift. These Hilbert phase measures are designed to deal with the case of phase changing with time as in Fig 4e.

130. The capture of single cell data is not clear from this description.

134 Synchronization is not quantified. What kind of synchronization? Is it phase synchronization? Is it period locking?

171. How is alignment or misalignment quantified? How do the authors quantify the synchronization in Per2 under LH-1, LH-2, and LH-3? What is the phase variation in each experiment over time? The authors have stated they have single cell resolution. What are the percentiles (across cells) in the phase over time?

162. If the aim is to show entrainment, why are not oscillatory patterns to glucose and insulin being applied rather than a single pulse? Setting up a repeated pulse would be more informative and show true entrainment to the glucose signal. It might be useful to mention at this point that the periodic pulse is imposed later, and this whole setup is a control.

165. What does the periodic signal look like when the power spectra are computed on single cells using the 30 m resolution data? Fig 1B is not sufficient. How do the authors handle the cells settling into a stable limit cycle? Why is the luminescent response damping out? What experiments have been done to explain the damping?

167. Why does the 12:12 give the most sustained pulse? How does this relate to the early work of Pittendrigh and other findings at the single cell level in clock systems?

206. There is a conceptual gap here. The transcriptomic results are being done on a macroscopic scale and being compared with single cell results. Why do the authors expect to see a correspondence? Why do the authors expect the microscopic results to recapitulate the macroscopic results? What happens when single cell transcriptomics are performed?

209 How do the trajectories of bioluminescence on Per2 at the single cell level compare with the Per2 trajectories by transcriptomics?

209 What happens to the cycles in a per2 knockout?

269. This statement is incorrect. The authors are simply not citing related work on single cell oscillatory perturbations that have been done on other model clock systems.

312. The phase analysis is just incomplete and needs to be more thorough with an examination of phase plotted as a function of time for single cells, for example. What do the percentiles and mean phase look like over time?

316. Where was this optimality result first reported?

335. It might be worth mentioning one of the author's own work and that of O'Donnell in the Royal Society on feeding and fasting and its impact on malaria as well.

343-6 It was mentioned that labview was used; however, the schematic that was shown did not show the full set up. A block scheme of the experimental set up maybe useful.

365. Very few details are given on the microfluidic chips. What are standard soft lithography techniques? How was the mask for the device designed? What was the PDMS substrate? How were the devices pretreated? Since the 2 devices were made from PDMS, how was the autofluorescence handled? The methods section is incomplete.

439. Image processing is not defined in sufficient detail to know what is being done. It would help to see what a blown up image looks like in a supplementary figure. For example, correction for cosmic ray artifacts means nothing to the reader. The image processing steps from source to detector need to be layed out. How were single cells identified? If a workflow in MATLAB was developed, it needs to put in GitHub.

439. Where is the single cell data deposited?

443 Adjusting the brightness and contrast would alter the results obtained, giving false results if this was utilized during the image processing step.

445. ROI is not defined, although a region of interest could mean anything from a FOV to the region around a cell.

465. How was the quality of the FASTQ files assessed? Was their quality assessed using Ewels MultiQC? What are some of the quality plots graphed for different time points? Will these be added to the supplement?

642 Microfluidic device schematic representation: was the culture chamber placed on a board of some sort? What are the long serpentine areas? Is it tubing or the microfluidic device? Needs to be more specific.

646. The authors are not presenting enough details on the microfluidic device (the schematic is ambiguous; were the devices pretreated?), and they are not making full use of its measurement capability. Why take measurements only every 6 h in Fig 1E? Why not every 30 m? If continuous observations are being taken as reported in the methods (i.e., 30 m intervals), why is the luminescence not being reported on a finer scale?

647 The luminescence data goes to negative; what methods for image processing, data cleaning or manipulation was done to get the results? This was not mentioned in the image processing step.

679. Nonparametric test is needed.

Reviewer #2:

Remarks to the Author:

The authors describe here the establishment and testing of a microfluidic device for extended tissue culture conditions and the manipulation of culture conditions. They use this device to optimize the synchronization of rhythms in cultured cells. Depending on the experimental set up, different cyclic output was observed. Overall, the study is well performed and well analysed.

In essence, the microfluidic system offers many experimental advantages. However, the biological information described in this paper is very limited, because the findings were already obtained in other tissue culture systems and even animals.

From my point of view, one crucial experiment is missing. A dexamethasone shock induces free-running, circadian oscillations. Does the relatively nonphysiological glucose/insulin treatment (in terms of 'daily' concentration changes) provoke free-running rhythms, or zeitgeber-dependent rhythms? To distinguish these possibilities, a circadian mutation has to be introduced changing the free-running period by a couple of hours. Either one would observe this change in period length in the transcriptome data, or the bulk of the data remains unchanged, which indicates the Zeitgeber's influence.

Reviewer #3:

None

Reviewer #1

The microfluidics approach is very interesting; however, the work is not tied into the clock literature, either experimental or theoretical.

We thank the reviewer for his/her appreciation, we now better contextualized our work within the literature. The following references have been added in the introduction and discussion sections:

Sato, M., Murakami, M., Node, K., Matsumura, R. & Akashi, M. The Role of the Endocrine System in Feeding-Induced Tissue-Specific Circadian Entrainment. *Cell Rep.* **8**, 393–401 (2014).

Damiola, F. *et al.* Restricted feeding uncouples circadian oscillators in peripheral tissues from the central pacemaker in the suprachiasmatic nucleus. *Genes Dev.* **14**, 2950–2961 (2000).

Stokkan, A. K. *et al.* Entrainment of the Circadian Clock in the Liver by Feeding Linked references are available on JSTOR for this article : Entrainment of the Circadian Clock in the Liver by Feeding. **291**, 490–493 (2001).

López-Otín, C., Galluzzi, L., Freije, J. M. P., Madeo, F. & Kroemer, G. Metabolic Control of Longevity. *Cell* **166**, 802–821 (2016).

Longo, V. D. & Panda, S. Fasting, Circadian Rhythms, and Time-Restricted Feeding in Healthy Lifespan. *Cell Metab.* **23**, 1048–1059 (2016).

The two parts of the paper are not logically connected because one part is on the microscopic scale of single cells, and the other part is on the macroscopic scale through standard RNA transcriptomics.

Microfluidics is a quite new approach in circadian studies, only few works previously explored the potential of this technology. Given our long-term experience on different strategies to perform robust cell culture in microfluidic (for instance, Giulitti *et al.*, Optimal periodic perfusion strategy for robust long-term microfluidic cell culture, Lab on a chip, 2013), we carefully evaluate the potential drawbacks in using microfluidic technology for circadian study. In particular, to avoid any artefact coming from the intrinsic properties of microfluidic setup, including small volume, the high surface to volume ratio and upstream and downstream properties created by unidirectional medium flow. We thought it was important to visualize the circadian behaviour in different region of the microfluidic channel and to capture any spatial heterogeneity along the channel and at the edges.

In the manuscript we had erroneously written that we performed luminescence experimental measurements with single-cell resolution. However, despite we used an imaging resolution of 20 microns, we did image analysis on a pixel basis, not cell basis. We did not perform any single-cell identification and single-cell tracking.

Thus, we now made clear this concept within the text and explain that high resolution was used for investigating the spatial heterogeneity of luminescence signals within microfluids. For avoiding misinterpretation, any reference to single-cell measurement were eliminated.

We thank the reviewer for all suggestions about the single-cell analysis, which avoids misunderstanding and, in the revised manuscript, the analysis of Per2 oscillations and the transcriptomics sounds better connected.

We anticipate that this clarification will probably overcome many of the issues raised by the reviewer about the proper use of single-cell analysis.

In the next answers, we will refer to the explanation given at this point.

The authors do not make use of the continuous time measurement at the single cell level.

As mentioned above, we agree with the reviewer that we cannot state that the measurement was performed at single-cell level, because we did not track cells. Instead, we resolved the spatial heterogeneity of luminescence signal throughout the microfluidic channel surface. Our data show that a circadian oscillatory pattern can be established quite homogeneously throughout this surface, and edge effect or upstream and downstream effects are not observed.

We added this piece of information in the manuscript (lines 110-112) and clarified that we used high-resolution imaging coupled with microfluidic technology for the circadian study.

The description of the microfluidics needs to be more complete in Materials and Methods.

We thank the reviewer for highlighting this weakness of the manuscript. We added more details in Methods section (Microfluidic Devices- lines 392-422) and in the supplementary figure (Fig. S1) a more detailed scheme describes the microfluidic platforms used.

They do not characterize the level of detection noise in their single cell measurements, leaving in question the validity of results presented in the microfluidics experiments. The phase analysis is incomplete and not contemporary.

Also this comment falls within the single-cell measurement misunderstanding. Our main results are based on an imaging analysis on ROIs of 8×8 pixels, where the pixel size is 20 microns, according to the methods reported in Material and Methods (Image Processing). This ROI size was chosen as a trade-off between spatial heterogeneity and robustness of measurement, according to measurement noise evaluation, as pointed out by the reviewer. The signals integrated over the single pixel in the defined ROI is high and reproducible compare to the background.

These high-resolution measurements, reported in Fig. 1, provide evidence for a consistent cellular response throughout the entire surface of the culture chamber.

Manuscript has been modified accordingly.

The connection to existing models of the clock is absent. For example, if the cells constitute damped oscillators, then one might argue they can entrain to any period of the driving input. On the other hand if single cells have their own intrinsic clock, they can entrain only within a range of frequencies about the intrinsic frequency of each cell. From the data presented it is not clear that single peripheral cells have a clock. Not knowing that individual cells have clocks leaves open how to explain the entrainment phenomenon observed.

This question is based on the idea that we explored the Per2 expression of the cells by a single cell measurement. However, after system validation in Figure 1 using high-spatial resolution, we performed all our experiments using a bulk analysis approach. Figure 1 will be important to assess the capability of our platform to be useful for the circadian study we have in mind, and all the important findings achieved are given by a bulk analysis of a single microfluidic channel.

However, the reviewer is asking important question if the input is able to entrain the intrinsic cell clock or not. All single cells show a cell-autonomous and self-sustained circadian oscillation with undiminished amplitude and diverse circadian periods and this circadian expression continues also during cell division as previously demonstrated (Nagoshi, E. *et al.*

Circadian gene expression in individual fibroblasts: Cell-autonomous and self-sustained oscillators pass time to daughter cells. *Cell* **119**, 693–705 (2004); Welsh, D. K., Yoo, S. H., Liu, A. C., Takahashi, J. S. & Kay, S. A. Bioluminescence imaging of individual fibroblasts reveals persistent, independently phased circadian rhythms of clock gene expression. *Curr. Biol.* **14**, 2289–2295 (2004)). Cells can be partially synchronized in phase by different agents (e.g. serum shock, dexamethasone) at the population level, however, lack of oscillator coupling in cell cultures leads to a loss of synchrony among individual cells and damping of the ensemble rhythm. Bioluminescence imaging of individual fibroblasts reveals persistent, independently phased circadian rhythms of clock gene expression. Thus, the Per2 and transcriptomics oscillations, we observed, is always based on entrainment of the entire cellular population.

Line_number

43. What is missing in the introduction is a relation of the work to other model systems on the clock, such as in Arabidopsis, where glucose entrainment was reported earlier in Nature. What is also missing is the first global descriptions of the clock network in other model systems to help interpret the transcriptomics data.

Thanks for this comment, we revised the introduction and added this ref. n. 7 and 8 lines n.51-52.

80. The microfluidic approach is to be commended, but their microfluidic approach to the entrainment problem is not novel and has been utilized in other model clock systems, but the authors cite none of this work. only 1 paper is missing.

Thanks for this comment, we added the most relevant papers we found (lines n. 91-96) and updated the bibliography accordingly (ref. 39-42).

104. This measurement of clock gene luminescence in single cells in an entrainment experiment is not the first time this has been done. This has been done in other clock systems, but again it would be useful to relate the results here to work on other clock systems. For example, how did the authors go about separating the detection and stochastic intracellular noise in the luminescent measurements made?

We did not perform any single cell analysis. We always refer to published protocol for analysing bulk signals from cell population.

Single cell analysis would have required a different approach, as the one we published recently (Li, Y. *et al.* Noise-driven cellular heterogeneity in circadian periodicity. *Proc. Natl. Acad. Sci. U. S. A.* **117**, 10350–10356, 2020. Li, Y. *et al.* Epigenetic inheritance of circadian period in clonal cells. *Elife* **9**, 1–36, 2020).

105 What fraction of the variation in luminescence is detection noise? How is the reproducibility of the measurements quantified? What controls have been done with luminescent beads substituted for living cells and the like? Without the characterization of the detection noise the microfluidic results remain in question.

We obtained the video presentation by putting the images acquired every 30 min in sequence using ImageJ; cosmic ray artifacts were removed by ImageJ (function Process>Noise> Remove Outliers) as already reported in our previous publications (Li, Y. *et al.* *Proc. Natl. Acad. Sci. U. S. A.* **117**, 10350–10356, 2020; Li, Y. *et al.* *Elife* **9**, 1–36, 2020).

We designed manually ROIs of the channels and measured the luminescence signal of each by using ImageJ (function Image>Stacks> Plot Z-axis Profile).

For the luminescence data that goes to negative, we used also LumiCycle software to subtract the baseline.

All these procedures are added in the Methods section (Image processing) and relative references have been added.

107 The authors have not taken advantage of this system and shown continuous data acquisition. Why have they not done so, if the systems is so reproducible? How do the findings on *Per2::Luc* compare with homologs in other systems, where single cell measurements have been taken on the clock mechanism?

In view of guaranteeing reproducibility, the continuous acquisition of luminescence signal was integrated over 30-minute intervals to increase signal-to-noise ratio. The phrase n. 107 has been removed and the details are better explained.

108. They authors assert high resolution at single cell level. How were cells identified?

130. The capture of single cell data is not clear from this description.

As mentioned before, we have performed high-spatial resolution measurements with a 20 microns resolution (cell size), although a single-cell analysis has not been performed.

127. The authors are not using reasonable and well studied measures of phase, such as the Hilbert phase. These contemporary phase measures like Hilbert phase allow one to get away from assuming constant phase with the phase shift. These Hilbert phase measures are designed to deal with the case of phase changing with time as in Fig 4e.

Considering the line 127, and the Fig. S2c, we measured the phase shift as time difference between two corresponding peaks of the two media changes implemented as schematized in Fig. S2a.

We agree with the reviewer that more accurate estimations of the phase exist. However, RNA-seq data were acquired for the time span of a single cycle. The phase was determined using the quite established algorithm JTK. This phase determination is just a rough estimation: as shown in Fig. 4e, the phase is determined with 2h time resolution. Our biological conclusions are based on large phase differences. We agree that we could have missed more subtle transcriptional differences, but such results would also require a different experimental design with RNA sampling for a much longer period of time.

134 Synchronization is not quantified. What kind of synchronization? Is it phase synchronization? Is it period locking?

Our study is focused on how repeated stimulations of feeding and fasting cycles are able to synchronize the phase of *Per2* of the cell population.

Analysis of the period was initially excluded from the analysis, excepted for the first characterization in the microfluidics use as shown in Fig. S2b. However, we did an additional experiment reported in Fig. 3e-f to better understand if our stimulus of glucose/insulin applied at different cycle times (20h, 24h, 26h) provides free-running rhythms, or zeitgeber-dependent rhythms. We found that the stimulation of feeding (HIGH) and fasting (LOW) cycles of 20h and 26h significantly change the period of *Per2* expression compare to 24h cycles.

Comments and discussion about this new experiment have been added.

171. How is alignment or misalignment quantified? How do the authors quantify the synchronization in Per2 under LH-1, LH-2, and LH-3? What is the phase variation in each experiment over time? The authors have stated they have single cell resolution. What are the percentiles (across cells) in the phase over time?

The alignment and the misalignment depend on the coherence between the time of when we stimulate the cells, that always corresponds to the trough of Per2 expression and the type of the stimulus applied, HIGH (feeding) or LOW (starvation). We better specified this concept in the manuscript.

Considering that the activity phase (when glucose and insulin are high) occurs between the minimum and the maximum of Per2, and the rest activity (when glucose and insulin are low) occurs between the maximum and the minimum of Per2, HL mimics an aligned condition since the HIGH Stimulus is applied coherently with the phase activity. On the other hand, LH mimics a misaligned condition since the LOW Stimulus is applied incoherently with the phase activity. The phase variation is the time difference between two corresponding peaks of the two experimental conditions compared as we explained previously. The variability of each experiment has now been evaluated and reported in the captions.

As already explained in the previous questions, we did not do any single cell resolution analysis but only bulk.

162. If the aim is to show entrainment, why are not oscillatory patterns to glucose and insulin being applied rather than a single pulse? Setting up a repeated pulse would be more informative and show true entrainment to the glucose signal. It might be useful to mention at this point that the periodic pulse is imposed later, and this whole setup is a control.

The experiment reported in Fig. 2a-b aimed at deciding the time and the frequency at which we switch between H (feeding) and L (fasting) media and vice versa within a 24h cycle, among these combinations: 4:20h, 8:16h, 12h:12h, 16:8h and 20:4h. Once we decided that 12h:12h better sustains the Per2 oscillations, we did repeated HIGH and LOW stimulations up to 3 days.

We rephrase the related text to make clearer the idea that only aligned 24-hour HL cycles are able to synchronize the cell population and provides strong entrainment.

165. What does the periodic signal look like when the power spectra are computed on single cells using the 30 m resolution data? Fig 1B is not sufficient. How do the authors handle the cells settling into a stable limit cycle? Why is the luminescent response damping out? What experiments have been done to explain the damping?

As mentioned above, reported data are not single-cell data, and the subsequent questions are still related to this aspect. Signal damping is due to desynchronization as previously reported, not to single-cell damping.

Indeed, during the Free Running acquisition, the luminescence damping occurs because of progressive phase desynchronization between cells belonging to the whole culture (Nagoshi et al., 2004; Welsh et al., 2004). While single-cell luciferase emission in fibroblasts remains high throughout and self-sustained (Nagoshi et al., 2004; Welsh et al., 2004, Li Y. et al 2020), the whole culture undergoes toward a desynchronization due to the absence of intercellular signaling coupled arising for example from cell division (Nagoshi et al., 2004).

On the other hand, what we observed in our experiments is that 12/12 HL (for 1, 2 3 cycles) matches the phase of the Per2 oscillation and no damping was observed, as it is possible to appreciate in Fig. 3a-b.

About the Figure 1b, we added Figure S1 to increase the details of the microfluidic platforms used.

167. Why does the 12:12 give the most sustained pulse? How does this relate to the early work of Pittendrigh and other findings at the single cell level in clock systems?

We analysed the overall cell population, in which the fibroblast cell period is heterogeneous, the most common period of Per2 is between 24 and 25h (Tanya L. Leise, Persistent Cell-Autonomous Circadian Oscillations in Fibroblasts Revealed by Six-Week Single-Cell Imaging of PER2::LUC Bioluminescence, Plos one 2012; David K. Welsh, Bioluminescence Imaging of Individual Fibroblasts Reveals Persistent, Independently Phased Circadian Rhythms of Clock Gene Expression, Current Biology 2004). Our data showed that using 12/12 HL and 24h cycle, we were able to match the intrinsic clock of the cells and obtain sustained signals.

We discussed that the interplay between the frequency of the metabolic entrainment and the peripheral circadian clock need to be aligned.

The effectiveness of 12:12 may indeed relate to Pittendrigh's hypothesis of "resonance" during entrainment in which the amplitude of the circadian oscillator in enhance when the driving entrainment signal and the underlying circadian clock are in resonance (i.e., have similar period values).

Although deeply understanding of why the resonance hypothesis seems to be interesting, the study of specific molecular mechanisms is beyond the aim of this study.

206. There is a conceptual gap here. The transcriptomic results are being done on a macroscopic scale and being compared with single cell results. Why do the authors expect to see a correspondence? Why do the authors expect the microscopic results to recapitulate the macroscopic results? What happens when single cell transcriptomics are performed?

This comment, as the others, is related to the single-cell circadian analysis. We performed luminescent imaging acquisition with high spatial resolution to set-up the entire experimental microfluidic system, but we did not tract the single cells. Our data are always analysed at population level.

209 How do the trajectories of bioluminescence on Per2 at the single cell level compare with the Per2 trajectories by transcriptomics?

We have not tracked the single cells, but we made a measurement of bioluminescent signal at the population level. For this reason, since we always analysed a population behavior rather than a single-cell one, it is possible to compare all our data together.

Transcriptomic data have the advantage of simultaneous detection of many transcripts, at the price of a lower signal-to-noise ratio. Per2 was found to significantly oscillate only in HL condition, while other genes of the core clock were found to oscillate in all 3 conditions (HL, LH, HH), see Fig. 4g.

209 What happens to the cycles in a per2 knockout?

Thanks for this interesting comment. We aim to investigate the effects that a wrong schedule of habits (e.g. feeding and fasting) on a healthy clock. We add a phrase on the discussion about these possibilities.

269. This statement is incorrect. The authors are simply not citing related work on single cell oscillatory perturbations that have been done on other model clock systems.

Thanks for this comment, we added the most relevant papers we found (lines n. 91-96) and updated the bibliography accordingly (ref. 39-42).

312. The phase analysis is just incomplete and needs to be more thorough with an examination of phase plotted as a function of time for single cells, for example. What do the percentiles and mean phase look like over time?

We performed luminescent imaging acquisition at high spatial resolution only to set-up the entire experimental system, but we did not tract the single cells.

316. Where was this optimality result first reported?

Thanks for the suggestion, we better highlight where the optimal conditions were set.

335. It might be worth mentioning one of the author's own work and that of O'Donnell in the Royal Society on feeding and fasting and its impact on malaria as well.

Thanks for the suggestion, we added two references that can easily contextualized our work in a broader perspective.

343-6 It was mentioned that labview was used; however, the schematic that was shown did not show the full set up. A block scheme of the experimental set up maybe useful.

Thanks for this comment, we added citation of previous work and publicly provided the block scheme of the experimental set up in <https://github.com/Onelia-G/CAVRO-PUMPS-FOR-MICROFLUIDICS>.

365. Very few details are given on the microfluidic chips. What are standard soft lithography techniques? How was the mask for the device designed? What was the PDMS substrate? How were the devices pretreated? Since the 2 devices were made from PDMS, how was the autofluorescence handled? The methods section is incomplete.

Thanks for the comment, we improved the Material and Methods on the section of Microfluidic Devices and add more details in the Fig. S1.

About the autofluorescence, actually we use luminescence and not fluorescence, but even in case of luminescence PDMS does not provide any problem of autofluorescence as described in this work (The autofluorescence of plastic materials and chips measured under laser irradiation. Lab on a Chip 2005).

439. Image processing is not defined in sufficient detail to know what is being done. It would help to see what a blown up image looks like in a supplementary figure. For example, correction for cosmic ray artifacts means nothing to the reader. The image processing steps from source to detector need to be layed out. How were single cells identified? If a workflow in MATLAB was developed, it needs to put in GitHub.

We re-wrote the Image Processing in the Methods section, describing all the procedures with more details and the relative references have been added.

We obtained the video presentation by putting the images acquired every 30 min in sequence using ImageJ; cosmic ray artifacts were removed by ImageJ (function Process>Noise> Remove

Outliers) as already reported in our previous publications (Li, Y. *et al. Proc. Natl. Acad. Sci. U. S. A.* **117**, 10350–10356, 2020; Li, Y. *et al. Elife* **9**, 1–36, 2020).

We added our Matlab code for creating the heat map shown in Fig. 1g in GitHub and now is publicly available at this link <https://github.com/Onelia-G/heap-map>

439. Where is the single cell data deposited?

Bulk RNA-seq data presented in this study have been deposited at the Gene Expression Omnibus database (<https://www.ncbi.nlm.nih.gov/geo/query/acc.cgi?acc=GSE167763>) with the dataset identifier GSE167763 (token qvzvwuampjqrlyf).

443 Adjusting the brightness and contrast would alter the results obtained, giving false results if this was utilized during the image processing step.

Actually, we erroneously wrote that we adjusted the brightness and the contrast. What we wanted to say is that we removed the cosmic ray artifacts by pixelwise comparison of two consecutive images using the minimum value of each pixel using ImageJ, as reported in the work *Noise-driven cellular heterogeneity in circadian periodicity. PNAS 2020.*

445. ROI is not defined, although a region of interest could mean anything from a FOV to the region around a cell.

Thanks for the suggestion, we defined better the ROI at lines n. 494-502.

465. How was the quality of the FASTQ files assessed? Was their quality assessed using Ewels MultiQC? What are some of the quality plots graphed for different time points? Will these be added to the supplement?

The analysis of the quality of the samples was made through FastQC/MultiQC taking into account the general statistics of fastqc (Per base quality, Per sequence GC content, Overrepresented sequences). The overall report containing these information can be found here: <https://drive.google.com/file/d/1IVBSN8MjsXrepH8CfsQzJrwxYX9UXoc6/view?usp=sharing>.

We also addressed pre-processing quality by statistics related to alignment and quantification (amount of unique mapped reads on the total, counted reads ,detected genes), reported here:<https://drive.google.com/file/d/1oiUpegdwG3aRahgST6O6ANjZ47F0DFPn/view?usp=sharing>.

642 Microfluidic device schematic representation: was the culture chamber placed on a board of some sort? What are the long serpentine areas? Is it tubing or the microfluidic device? Needs to be more specific.

Thanks for this comment, we provided more details about the microfluidic platforms used in the Material and Methods (Microfluidic Device) and also added the Figure S1.

The culture chamber is placed on a borosilicate glass slide, placed in a petri dish and located on the microscope stage, as described in the Methods section; the serpentine is a part of the microfluidic chip as the Supplementary Fig. 1 shows.

646. The authors are not presenting enough details on the microfluidic device (the schematic is ambiguous; were the devices pretreated?), and they are not making full use of its measurement capability. Why take measurements only every 6 h in Fig 1E? Why not every 30 m? If continuous observations are being taken as reported in the methods (i.e., 30

m intervals), why is the luminescence not being reported on a finer scale?

We provided more details about the microfluidic platforms used in the Methods (Microfluidic Devices) and also added the Fig. S1.

Fig. 1e shows Real-Time PCR data of human fibroblasts harvested every 4 h for 48h. This technique quantifies the expression level of one or more genes of interest of the sample harvested at a precise time point. The frequency of sample collection depends on an operator who for obvious reasons cannot harvest the cells every 30 min. However, 4h interval allows to follow the temporal dynamics of a circadian gene.

Different considerations need to do for the luminescence measurement, that is based on the acquisition under the microscope of a luminescent signal expressed from a luciferase-reporter cell line. With this approach it is possible to implement a frequency of measurement with a high temporal resolution, every 30 min, with a good quality of imaging, as we described previously; on the other hand, it possible to follow the expression of only one gene.

647 The luminescence data goes to negative; what methods for image processing, data cleaning or manipulation was done to get the results? This was not mentioned in the image processing step.

To analyze the circadian parameters, we first need to remove baseline changes because there are often drastic changes in the baseline in the first few days in culture. The baseline drift was obtained by fitting a polynomial curve by LumiCycle Software (Li, Y. *et al.* Epigenetic inheritance of circadian period in clonal cells. *Elife* **9**, 1–36, 2020; Li, Y. *et al.* Noise-driven cellular heterogeneity in circadian periodicity. *Proc. Natl. Acad. Sci. U. S. A.* **117**, 10350–10356, 2020; Yamazaki, S. & Takahashi, J. S. Real-time luminescence reporting of circadian gene expression in mammals. *Methods in Enzymology* **393**, 288–301, 2005).

679. Nonparametric test is needed.

We acknowledge that, for temporal series, data collected within the same data series are not iid (independent and identically distributed), however we applied t-test and ANOVA on parameters, like phase and frequency, to detect differences between independent experimental conditions, thus it is reasonable to use these tests.

Reviewer #2

The authors describe here the establishment and testing of a microfluidic device for extended tissue culture conditions and the manipulation of culture conditions. They use this device to optimize the synchronization of rhythms in cultured cells. Depending on the experimental set up, different cyclic output was observed. Overall, the study is well performed and well analysed.

We would like to thank the reviewer for carefully reading our work and for his positive words.

In essence, the microfluidic system offers many experimental advantages. However, the biological information described in this paper is very limited, because the findings were already obtained in other tissue culture systems and even animals.

In the recent years, advances in microscale manufacturing methods have enabled the generation of dynamic cell culture systems to recapitulate capture microenvironments with physiological relevance. Micro-physiological systems (MPSs) are microfluidic cell culture platforms capable of enabling the development of reproducible, physiologically relevant culture environments necessary to support the characteristic functionality of the primary tissue of origin over an extended culture period. These systems, which are often referred to as “organs-on-a-chip” or “tissues-on-a-chip,” combine advances in microfabrication and microfluidics technology to present highly controlled extracellular cues to cells in a physiologically relevant context.

Conventional cell culture systems are static, and their design precludes the biological complexity that is required to model dynamic changes characteristic of real system.

Given that, the microfluidic cell culture system offers a unique possibility to perform defined temporal resolved perturbations while recording cellular functions. In addition, as an in vitro model, it would be possible, as we addressed in our study, to dissect specific cyclic perturbation on peripheral circadian clock from other signals. This will be extremely difficult in animal studies where it would be impossible to distinguish metabolic fluctuations from systemic other signals.

We added specific comment along the introduction.

From my point of view, one crucial experiment is missing. A dexamethasone shock induces free-running, circadian oscillations. Does the relatively nonphysiological glucose/insulin treatment (in terms of ‘daily’ concentration changes) provoke free-running rhythms, or zeitgeber-dependent rhythms? To distinguish these possibilities, a circadian mutation has to be introduced changing the free-running period by a couple of hours. Either one would observe this change in period length in the transcriptome data, or the bulk of the data remains unchanged, which indicates the Zeitgeber’s influence.

We thank the reviewer for emphasizing the important question if the non-physiological glucose/insulin treatment provokes free-running rhythms, or zeitgeber-dependent rhythms. This is an important question that was missing from our first submission.

To address this question, we envision to develop a number of different strategies. On one side, as suggested by the reviewer, it would be possible to introduce a circadian mutation that will shift the free-running period by a couple of hours. Unfortunately, we do not have availability of cell line with mutation that affects period of circadian oscillation. Moreover, the cell population circadian is determined by averaging widely dispersed periods of individual clock cells.

On the other hand, in spite of changing the intrinsic cellular circadian clock, we hypothesize that using the same cell line as in all other experiments, without affecting the circadian genetic circuit, we could apply cyclic perturbation with different periods. In addition, this last strategy will allow to compare the results obtained with shorten or longer period perturbation with all other conditions test in our submitted work.

Thus, we designed a new experiment where we performed cyclic feeding and fasting perturbations (HL) applied at different period of 20, 24 and 26 hours. As control of circadian rhythm, we used dexamethasone and, as control of feeding/fasting cycles we performed HH perturbations. All experiments were performed in parallel for 3 cycles and Per2 were analyzed in the free running conditions.

The results showed that HL 12/12 h aligned with the intrinsic period of the cells (24h) showed sustained circadian oscillations with a period of 24h. Similar period of Per2 oscillation was detected for all controls, including dexamethasone and HH with different periods of stimulation, that are in absence of the fasting phase. Interestingly, HL cyclic perturbations with period of 20h and 26h showed alteration of Per2 oscillatory patterns with a longer period. However, the Per2 oscillation of both HL of 20 and 26 hours are damped in the free running. This result is similar to those observed in misaligned perturbation or feeding/fasting cycles far from 12/12.

From this experiment we can conclude that 12/12 hours HL sustains oscillations aligned with the intrinsic peripheral clock.

We added these new experiments in Fig. 3e-f. Description of the results and discussion were added in the revised manuscript.

According to this experimental evidence, we decide to not perform RNAseq analysis in different perturbation periods (20h and 26h) because the disruption of Per2 oscillations likely could lead to an obvious result: transcriptome will show random oscillatory patterns.

Reviewer #3

The manuscript presents interesting work on using a microfluidic platform to monitor circadian oscillations (recording circadian gene expression of modified fibroblast cells through luminescence detectors) over an extended time. The topic is hot and may be of interest to many readers. This paper opens a field, where similar microfluidic platforms may be developed to monitor specific cellular disease models, in response to drug treatment, by modeling typical circadian clock perturbations linked to particular lifestyle choices. This addition could lead to a new, multidimensional, and perhaps more personalized disease models.

We thank the reviewer for sharing his enthusiasm about our work. We add few more sentences at the end of the discussion to describe possible implication of integrating a model of particular lifestyle in in vitro experimentation of circadian field.

In the introduction, the authors suggested that "many mammalian behaviors and physiology processes, such as sleeping and feeding, are regulated during the 24h solar day by the circadian clock system". Although true, this statement requires confirmation in the cited literature, perhaps a review paper on the topic, so that the readers who are less familiar with the subject can refer to (line 43-44).

Thanks for this suggestion we added some interesting references in the lines considered.

As the authors state, continuing desynchronization between physiological and behavioral rhythms, such as shift work, sleep disruption, and abnormal feeding schedules,) may carry a significant risk of various disorders, ranging from sleep disorders to diabetes, obesity, cardiovascular disease, and cancer. Desynchronization of food intake from the circadian clock system in humans is an essential factor in the onset of metabolic disorders, proving that the paper's main topic is very timely and important.

As the authors suggest, "a deeper understanding of the mechanisms by which environmental inputs affect the circadian clock and rhythmicity of cellular functions is critically important for preventing diseases."

We thank the reviewer to fully understand the full potential of our work. According to other reviewer comments we re-wrote some sentences to make this concept even more clear.

Unfortunately, designing an experimental strategy for investigating and dissecting the contribution of a specific oscillatory metabolic pattern on the circadian clock is challenging. The main reasons are explained briefly in the manuscript.

Thank you for the question which allows us to explain further our strategy. We made clearer our motivations and rational, as you can appreciate in the introduction.

Because of the challenges, the authors took advantage of microfluidic technology to perform periodic cyclic stimulations under controlled microenvironment conditions while continuously monitoring circadian oscillations (recording circadian gene expression through luminescence detectors). The microfluidic device successfully allowed the authors to perform frequency-encoded metabolic stimulation on the cell-autonomous circadian clock to alter the cellular transcriptome's rhythmicity.

To meet the reviewer comments and appreciations, we added in the manuscript schematic diagram of the microfluidic setup, photomask file ready to use to reproduce the chip by

standard soft lithography, and detailed procedure to perform similar experiment without having a specialized background.

We envision that in the future, microfluidic technology could become an important tool in the circadian field and more in general in time dependent study of the cellular biology.

In the results section, the authors described the "microfluidic technology for circadian studies." The so-called microfluidic platform was designed by the authors to allow recording circadian gene expression by luminescence detectors while performing frequency-encoded metabolic perturbations on Per2::Luc mouse fibroblasts cultured inside a 2-d microfluidic observation chamber. In this section (titled "microfluidic technology for circadian studies"), the authors described their specific microfluidic platform, in one sentence (line 99 of the manuscript). The authors did not provide clear explanations why the microfluidic platform is better over other methods (more static, like culture and monitoring in well plates, that can also be automated to perform long term image-based observations of the cell's behavior). Is the microfluidic platform better just because it allows for working with minimal volumes of culture medium, or allows rapid shifting of different environmental conditions to which the cells are exposed through a different culture medium, or it is relatively easy to automate?

On the other hand, the authors presented convincing results of the proof-of-concept that the microfluidic setup, automated with the software operating liquid handling equipment, can monitor luminescence signal from Per2::Luc fibroblasts in a microfluidic chamber at single-cell resolution.

We thank for highlighting the lack of full description of microfluidic platforms. We added a detailed description of microfluidic platforms in the first part of the results.

Figure S1 reporting the drawing and details about the microfluidic experimental setup.

We also added comments on specific benefits of using microfluidic setup in the circadian experiments. For instance, the laminar flow, low volume associated with rapid media changes, and predictable condition, make this system particularly controlled and useful.

These characteristics are also exploited in all the field of micro-physiological systems.

The authors suggested that the ability of their microfluidic system to manipulate and control the cellular microenvironments acquired by using external liquid handling offers unlimited cell culture perfusion strategies. The system is based on a multi-port syringe pump (Tecan, the used model is not mentioned in the document's method part, under microfluidic devices section).

We added more information about the external liquid handling in the Method part in the Microfluidic Devices section.

The manuscript, in the current form, lacks a good of why the authors decided to develop the microfluidic platform, except for mentioning that the "microfluidics has been used for dynamic studies in different applications, including the circadian field for studying the molecular mechanisms of circadian gene oscillations in algae and fungi, allowing long-term measurements with high spatial and temporal resolution." Adding that explanation to the main text would help the readers to understand the author's decision on why the microfluidic approach is superior to alternative methods. The manuscript is also not providing a clear explanation of whether there are any alternative ways that similar

experiments can be done, as well as if they have tried them, but the microfluidic platform is simply unbeatable.

A better description of microfluidic has been added in the context of state of the art for performing dynamic perturbations (lines 107-112, 162-168 and 392-422) and can help to understand the rationale of the use of microfluidics for studying circadian clock.

The microfluidic part of the manuscript is followed by sections in which the authors explore different conditions of the experimental setup, such as the frequency and timing of feeding-fasting cycle, and how the perfusion frequency affects the circadian clock, as well as how the number of cycles of feeding-fasting influences the cell-autonomous clock, also how the synchronization between the feeding-fasting cycles and the cell-autonomous clock (phase of Per2) affects the entire transcriptome.

It is worth to mention that we also added an additional experiment in which we modified the period of the perturbation ranging from 20 to 24 and 26 hours. The data showed that 12/12 hour feeding and fasting cycles of 24h provide sustained oscillatory pattern, in agreement with all other previous results.

In the discussion part of the manuscript, the authors mention that only a few papers were published on using microfluidics for studying the circadian clock (line 267). The cited studies were published in 2010 (ref. 33), 2013 (ref. 34 and 35). The authors comment that the papers were mainly limited to algae and fungi. A quick google search provides a more timely list, with just a few articles published on the topic listed below:

Single-cell in vivo imaging of cellular circadian oscillators in zebrafish Wang H, Yang Z, Li X, Huang D, Yu S, He J, et al. (2020) Single-cell in vivo imaging of cellular circadian oscillators in zebrafish. PLoS Biol 18(3): e3000435. <https://doi.org/10.1371/journal.pbio.3000435>

A microfluidic approach for experimentally modelling the intercellular coupling system of a mammalian circadian clock at single-cell level, Lab Chip, 2020,20, 1204-1211, <https://pubs.rsc.org/en/content/articlelanding/2020/LC/D0LC00140F#!divAbstract>

Thanks for this comment, we added the most relevant papers we found (lines n. 91-96) and updated the bibliography accordingly (ref. 39-42).

Overall the paper is relatively easy to follow, and the presented topic is fascinating and may open many scientific opportunities in the future in different fields. For example, one can imagine novel disease models that take into account modeling and monitoring circadian oscillations on a single-cell level for precision medicine drug screenings or monitoring circadian oscillations in 3-d cell cultures.

After providing minor changes to the manuscript (please check the comments and suggestions mentioned above), this work should be published in Nature Communications.

We thank once more the reviewer for sharing with us his enthusiasm.

Reviewers' Comments:

Reviewer #1:

Remarks to the Author:

General Comments by reviewer 1: Recognizing that single cell measurements are not being made is a good change, and accurately describing the microfluidic measurements over 8 x 8 pixel grid is an improvement, but it still does not remove the criticisms of reviewer 1. The questions still remain about the level of detection noise and stochastic intracellular noise on this 8 x 8 pixel scale. Approaches to addressing this problem of noise are published in the Nature family of journals on the clock. Moreover, the authors still need to cite the relevant literature in Nature Publisher family of journals and other journals where single cell measurements have been made in model systems to examine the entrainment question. What is being plotted in terms of bioluminescence is still not made clear. The phase analysis is not contemporary. For example, the phase plots in Figure 3c assume a constant phase, and this is probably not the case. This paper is not ready for publication because of shortcomings in the controls for the microfluidics experiments and the analysis of bioluminescence.

Below each section has the comment of reviewer 1 followed by the authors' response. Reviewer 1 then comments on the authors response as part of the current review. I have also taken the liberty to upload a pdf where the three different sections are bolded.

Reviewer #1

The microfluidics approach is very interesting; however, the work is not tied into the clock literature, either experimental or theoretical.

We thank the reviewer for his/her appreciation, we now better contextualized our work within the literature. The following references have been added in the introduction and discussion sections: Sato, M., Murakami, M., Node, K., Matsumura, R. & Akashi, M. The Role of the Endocrine System in Feeding-Induced Tissue-Specific Circadian Entrainment. *Cell Rep.* 8, 393–401 (2014).

Damiola, F. et al. Restricted feeding uncouples circadian oscillators in peripheral tissues from the central pacemaker in the suprachiasmatic nucleus. *Genes Dev.* 14, 2950–2961 (2000).

Stokkan, A. K. et al. Entrainment of the Circadian Clock in the Liver by Feeding Linked references are available on JSTOR for this article: Entrainment of the Circadian Clock in the Liver by Feeding. 291, 490–493 (2001).

López-Otín, C., Galluzzi, L., Freije, J. M. P., Madeo, F. & Kroemer, G. Metabolic Control of Longevity. *Cell* 166, 802–821 (2016).

Longo, V. D. & Panda, S. Fasting, Circadian Rhythms, and Time-Restricted Feeding in Healthy Lifespan. *Cell Metab.* 23, 1048–1059 (2016).

The two parts of the paper are not logically connected because one part is on the microscopic scale of single cells, and the other part is on the macroscopic scale through standard RNA transcriptomics.

Microfluidics is a quite new approach in circadian studies, only few works previously explored the potential of this technology. Given our long-term experience on different strategies to perform robust cell culture in microfluidic (for instance, Giulitti et al., Optimal periodic perfusion strategy for robust long-term microfluidic cell culture, *Lab on a chip*, 2013), we carefully evaluate the potential drawbacks in using microfluidic technology for circadian study. In particular, to avoid any artefact coming from the intrinsic properties of microfluidic setup, including small volume, the high surface to volume ratio and upstream and downstream properties created by unidirectional medium flow. We thought it was important to visualize the circadian behavior in different region of the microfluidic channel and to capture any spatial heterogeneity along the channel and at the edges.

In the manuscript we had erroneously written that we performed luminescence experimental measurements with single-cell resolution. However, despite we used an imaging resolution of 20 microns, we did image analysis on a pixel basis, not cell basis. We did not perform any single-cell identification and single-cell tracking.

Thus, we now made clear this concept within the text and explain that high resolution was used for investigating the spatial heterogeneity of luminescence signals within microfluids. For avoiding

misinterpretation, any reference to single-cell measurement were eliminated.

We thank the reviewer for all suggestions about the single-cell analysis, which avoids misunderstanding and, in the revised manuscript, the analysis of Per2 oscillations and the transcriptomics sounds better connected.

We anticipate that this clarification will probably overcome many of the issues raised by the reviewer about the proper use of single-cell analysis.

Comment by reviewer 1: In that so few papers have been published on microfluidics of the clock, it would be desirable and strongly suggested that the use of microfluidics in other model systems to examine light entrainment in single cells should be cited. There are a number of papers on this topic in Nature journals and other journals that are not cited and would provide insights, such as methods including controls and analyses, on the microfluidic data presented here.

In the next answers, we will refer to the explanation given at this point.

The authors do not make use of the continuous time measurement at the single cell level. As mentioned above, we agree with the reviewer that we cannot state that the measurement was performed at single-cell level, because we did not track cells. Instead, we resolved the spatial heterogeneity of luminescence signal throughout the microfluid channel surface. Our data show that a circadian oscillatory pattern can be established quite homogeneously throughout this surface, and edge effect or upstream and downstream effects are not observed. We added this piece of information in the manuscript (lines 110-112) and clarified that we used high-resolution imaging coupled with microfluidic technology for the circadian study.

Comment by reviewer 1: If there is this no spatial homogeneity in the signal, one should be able to demonstrate this by calculating the Kuramoto K on different locations (ROIs) or by plotting phase as a function of time for different ROIs on the microfluidic device surface.

The description of the microfluidics needs to be more complete in Materials and Methods. We thank the reviewer for highlighting this weakness of the manuscript. We added more details in Methods section (Microfluidic Devices- lines 392-422) and in the supplementary figure (Fig. S1) a more detailed scheme describes the microfluidic platforms used.

They do not characterize the level of detection noise in their single cell measurements, leaving in question the validity of results presented in the microfluidics experiments. The phase analysis is incomplete and not contemporary.

Also this comment falls within the single-cell measurement misunderstanding. Our main results are based on an imaging analysis on ROIs of 8×8 pixels, where the pixel size is 20 microns, according to the methods reported in Material and Methods (Image Processing). This ROI size was chosen as a trade-off between spatial heterogeneity and robustness of measurement, according to measurement noise evaluation, as pointed out by the reviewer. The signals integrated over the single pixel in the defined ROI is high and reproducible compare to the background.

These high-resolution measurements, reported in Fig. 1, provide evidence for a consistent cellular response throughout the entire surface of the culture chamber.

Manuscript has been modified accordingly.

Comment by the reviewer 1: There are published accounts in Nature journals to separate the detection noise from the stochastic intracellular noise (SIN) in other clock systems at the single cell level. In that the observations depend on characterizing the level of detection noise, a similar analysis is recommended here even if the observations (detection noise+ SIN) are averaged over 8x8 pixels. What happens to the signal if the area of integration is increased?

The connection to existing models of the clock is absent. For example, if the cells constitute damped oscillators, then one might argue they can entrain to any period of the driving input. On the other hand if single cells have their own intrinsic clock, they can entrain only within a range of frequencies about the intrinsic frequency of each cell. From the data presented it is not clear that single peripheral cells have a clock. Not knowing that individual cells have clocks leaves open how

to explain the entrainment phenomenon observed.

This question is based on the idea that we explored the Per2 expression of the cells by a single cell measurement. However, after system validation in Figure 1 using high-spatial resolution, we performed all our experiments using a bulk analysis approach. Figure 1 will be important to assess the capability of our platform to be useful for the circadian study we have in mind, and all the important findings achieved are given by a bulk analysis of a single microfluidic channel.

However, the reviewer is asking important question if the input is able to entrain the intrinsic cell clock or not. All single cells show a cell-autonomous and self-sustained circadian oscillation with undiminished amplitude and diverse circadian periods and this circadian expression continues also during cell division as previously demonstrated (Nagoshi, E. et al.

Circadian gene expression in individual fibroblasts: Cell-autonomous and self-sustained oscillators pass time to daughter cells. *Cell* 119, 693–705 (2004); Welsh, D. K., Yoo, S. H., Liu, A. C., Takahashi, J. S. & Kay, S. A. Bioluminescence imaging of individual fibroblasts reveals persistent, independently phased circadian rhythms of clock gene expression. *Curr. Biol.* 14, 2289–2295 (2004)). Cells can be partially synchronized in phase by different agents (e.g. serum shock, dexamethasone) at the population level, however, lack of oscillator coupling in cell cultures leads to a loss of synchrony among individual cells and damping of the ensemble rhythm.

Bioluminescence imaging of individual fibroblasts reveals persistent, independently phased circadian rhythms of clock gene expression.

Thus, the Per2 and transcriptomics oscillations, we observed, is always based on entrainment of the entire cellular population.

Comment by reviewer 1: With the microfluidics there is the opportunity to show that single cells can entrain to an external signal. The references above are limited to showing entrainment in bulk. It has already been demonstrated in other systems that single cells do entrain in other systems. Does it do so here at an intermediate scale of 8x8 pixels? The new results in Figure 3e-f would indicate a yes, but how does this compare with similar entrainment experiments done at the single cell in other model systems?

Line_number

43. What is missing in the introduction is a relation of the work to other model systems on the clock, such as in *Arabidopsis*, where glucose entrainment was reported earlier in *Nature*. What is also missing is the first global descriptions of the clock network in other model systems to help interpret the transcriptomics data.

Thanks for this comment, we revised the introduction and added this ref. n. 7 and 8 lines n.51- 52.

80. The microfluidic approach is to be commended, but their microfluidic approach to the entrainment problem is not novel and has been utilized in other model clock systems, but the authors cite none of this work. only 1 paper is missing.

Thanks for this comment, we added the most relevant papers we found (lines n. 91-96) and updated the bibliography accordingly (ref. 39-42).

Comments by Reviewer 1: Unfortunately the most relevant papers in *Nature* publications and other journals in related model systems are still not being cited.

104. This measurement of clock gene luminescence in single cells in an entrainment experiment is not the first time this has been done. This has been done in other clock systems, but again it would be useful to relate the results here to work on other clock systems. For example, how did the authors go about separating the detection and stochastic intracellular noise in the luminescent measurements made?

We did not perform any single cell analysis. We always refer to published protocol for analysing bulk signals from cell population.

Single cell analysis would have required a different approach, as the one we published recently (Li, Y. et al. Noise-driven cellular heterogeneity in circadian periodicity. *Proc. Natl. Acad. Sci. U. S. A.* 117, 10350–10356, 2020. Li, Y. et al. Epigenetic inheritance of circadian period in clonal cells. *Elife* 9, 1–36, 2020).

Comments by reviewer 1: The results reported here are intermediate between single cell analysis and bulk analysis. Averaging over 8x8 pixel grid still involves a new kind of analysis that takes account of both kinds of noise, detection noise and stochastic intracellular noise, at this level. Also a relevant control of following luminescent beads in a different chamber on the same device is not being used to examine the detection noise, for example.

105 What fraction of the variation in luminescence is detection noise? How is the reproducibility of the measurements quantified? What controls have been done with luminescent beads substituted for living cells and the like? Without the characterization of the detection noise the microfluidic results remain in question.

We obtained the video presentation by putting the images acquired every 30 min in sequence using ImageJ; cosmic ray artifacts were removed by ImageJ (function Process>Noise> Remove Outliers) as already reported in our previous publications (Li, Y. et al. Proc. Natl. Acad. Sci. U. S. A. 117, 10350–10356, 2020; Li, Y. et al. Elife 9, 1–36, 2020).

We designed manually ROIs of the channels and measured the luminescence signal of each by using ImageJ (function Image>Stacks> Plot Z-axis Profile).

For the luminescence data that goes to negative, we used also LumiCycle software to subtract the baseline.

All these procedures are added in the Methods section (Image processing) and relative references have been added.

Comments by reviewer 1: The video still doesn't constitute a control or a measurement of the detection noise. What about the controls such as luminescent beads in a different chamber on the same device? What does it mean to remove cosmic ray artifacts? How is this done? This needs to be described in the paper or its supplement. There needs to be a careful analysis of the detection noise.

107 The authors have not taken advantage of this system and shown continuous data acquisition. Why have they not done so, if the systems is so reproducible? How do the findings on Per2::Luc compare with homologs in other systems, where single cell measurements have been taken on the clock mechanism?

In view of guaranteeing reproducibility, the continuous acquisition of luminescence signal was integrated over 30-minute intervals to increase signal-to-noise ratio. The phrase n. 107 has been removed and the details are better explained.

Comments by reviewer 1. The homolog comparison done at the single cell level in another model system has not been done.

108. They authors assert high resolution at single cell level. How were cells identified?

130. The capture of single cell data is not clear from this description.

As mentioned before, we have performed high-spatial resolution measurements with a 20 microns resolution (cell size), although a single-cell analysis has not been performed.

127. The authors are not using reasonable and well studied measures of phase, such as the Hilbert phase. These contemporary phase measures like Hilbert phase allow one to get away from assuming constant phase with the phase shift. These Hilbert phase measures are designed to deal with the case of phase changing with time as in Fig 4e.

Considering the line 127, and the Fig. S2c, we measured the phase shift as time difference between two corresponding peaks of the two media changes implemented as schematized in Fig. S2a.

We agree with the reviewer that more accurate estimations of the phase exist. However, RNA-seq data were acquired for the time span of a single cycle. The phase was determined using the quite established algorithm JTK. This phase determination is just a rough estimation: as shown in Fig. 4e, the phase is determined with 2h time resolution. Our biological conclusions are based on large phase differences. We agree that we could have missed more subtle transcriptional differences, but such results would also require a different experimental design with RNA sampling for a much

longer period of time.

Comments by the reviewer 1: There are better ways to do the phase analysis. As an example the polar plot Figure 3c assumes constant phase, which is unlikely to hold in an entrainment experiment.

134 Synchronization is not quantified. What kind of synchronization? Is it phase synchronization? Is it period locking?

Our study is focused on how repeated stimulations of feeding and fasting cycles are able to synchronize the phase of Per2 of the cell population.

Analysis of the period was initially excluded from the analysis, excepted for the first characterization in the microfluidics use as shown in Fig. S2b. However, we did an additional experiment reported in Fig. 3e-f to better understand if our stimulus of glucose/insulin applied at different cycle times (20h, 24h, 26h) provides free-running rhythms, or zeitgeber- dependent rhythms. We found that the stimulation of feeding (HIGH) and fasting (LOW) cycles of 20h and 26h significantly change the period of Per2 expression compare to 24h cycles.

Comments and discussion about this new experiment have been added.

Comment by Reviewer 1: Thank you for the additional experiment. It would be very nice to overlay the driver in 3a-b on 3-d. Also a plot of phase of bioluminescence for LH-1, LH-2, and LH-3 as a function of time on the same graph would be good. This would help address how things are changing with regard to phase. Phase may be changing with time also, but this is not captured in the analysis. Also only by choosing the appropriate phase measure can you independently test for a period change and a phase change.

171. How is alignment or misalignment quantified? How do the authors quantify the synchronization in Per2 under LH-1, LH-2, and LH-3? What is the phase variation in each experiment over time? The authors have stated they have single cell resolution. What are the percentiles (across cells) in the phase over time?

The alignment and the misalignment depend on the coherence between the time of when we stimulate the cells, that always corresponds to the trough of Per2 expression and the type of the stimulus applied, HIGH (feeding) or LOW (starvation). We better specified this concept in the manuscript.

Considering that the activity phase (when glucose and insulin are high) occurs between the minimum and the maximum of Per2, and the rest activity (when glucose and insulin are low) occurs between the maximum and the minimum of Per2, HL mimics an aligned condition since the HIGH Stimulus is applied coherently with the phase activity. On the other hand, LH mimics a misaligned condition since the LOW Stimulus is applied incoherently with the phase activity. The phase variation is the time difference between two corresponding peaks of the two experimental conditions compared as we explained previously. The variability of each experiment has now been evaluated and reported in the captions.

As already explained in the previous questions, we did not do any single cell resolution analysis but only bulk.

Comments by reviewer 1: It is possible to measure the metabolites (glucose) simultaneously in these living cells now and needs to be tracked along with Per2.

162. If the aim is to show entrainment, why are not oscillatory patterns to glucose and insulin being applied rather than a single pulse? Setting up a repeated pulse would be more informative and show true entrainment to the glucose signal. It might be useful to mention at this point that the periodic pulse is imposed later, and this whole setup is a control.

The experiment reported in Fig. 2a-b aimed at deciding the time and the frequency at which we switch between H (feeding) and L (fasting) media and vice versa within a 24h cycle, among these combinations: 4:20h, 8:16h, 12h:12h, 16:8h and 20:4h. Once we decided that 12h:12h better sustains the Per2 oscillations, we did repeated HIGH and LOW stimulations up to 3 days.

We rephrase the related text to make clearer the idea that only aligned 24-hour HL cycles are able to synchronize the cell population and provides strong entrainment.

165. What does the periodic signal look like when the power spectra are computed on single cells using the 30 m resolution data? Fig 1B is not sufficient. How do the authors handle the cells settling into a stable limit cycle? Why is the luminescent response damping out? What experiments have been done to explain the damping?

As mentioned above, reported data are not single-cell data, and the subsequent questions are still related to this aspect. Signal damping is due to desynchronization as previously reported, not to single-cell damping.

Indeed, during the Free Running acquisition, the luminescence damping occurs because of progressive phase desynchronization between cells belonging to the whole culture (Nagoshi et al., 2004; Welsh et al., 2004). While single-cell luciferase emission in fibroblasts remains high throughout and self-sustained (Nagoshi et al., 2004; Welsh et al., 2004, Li Y. et al 2020), the whole culture undergoes toward a desynchronization due to the absence of intercellular signaling coupled arising for example from cell division (Nagoshi et al., 2004).

On the other hand, what we observed in our experiments is that 12/12 HL (for 1, 2 3 cycles) matches the phase of the Per2 oscillation and no damping was observed, as it is possible to appreciate in Fig. 3a-b.

About the Figure 1b, we added Figure S1 to increase the details of the microfluidic platforms used.

Comment by Reviewer 1: Even doing in bulk does not eliminate the problem of settling into a stable limit cycle. That problem persists at the macroscopic level as well or an intermediate level such as 8 x 8 pixel level.

167. Why does the 12:12 give the most sustained pulse? How does this relate to the early work of Pittendrigh and other findings at the single cell level in clock systems? We analysed the overall cell population, in which the fibroblast cell period is heterogeneous, the most common period of Per2 is between 24 and 25h (Tanya L. Leise, Persistent Cell- Autonomous Circadian Oscillations in Fibroblasts Revealed by Six-Week Single-Cell Imaging of PER2::LUC Bioluminescence, Plos one 2012; David K. Welsh, Bioluminescence Imaging of Individual Fibroblasts Reveals Persistent, Independently Phased Circadian Rhythms of Clock Gene Expression, Current Biology 2004). Our data showed that using 12/12 HL and 24h cycle, we were able to match the intrinsic clock of the cells and obtain sustained signals.

We discussed that the interplay between the frequency of the metabolic entrainment and the peripheral circadian clock need to be aligned.

The effectiveness of 12:12 may indeed relate to Pittendrigh's hypothesis of "resonance" during entrainment in which the amplitude of the circadian oscillator in enhance when the driving entrainment signal and the underlying circadian clock are in resonance (i.e., have similar period values).

Although deeply understanding of why the resonance hypothesis seems to be interesting, the study of specific molecular mechanisms is beyond the aim of this study.

206. There is a conceptual gap here. The transcriptomic results are being done on a macroscopic scale and being compared with single cell results. Why do the authors expect to see a correspondence? Why do the authors expect the microscopic results to recapitulate the macroscopic results? What happens when single cell transcriptomics are performed? This comment, as the others, is related to the single-cell circadian analysis. We performed luminescent imaging acquisition with high spatial resolution to set-up the entire experimental microfluidic system, but we did not tract the single cells. Our data are always analysed at population level.

Comment by Reviewer 1: The scale difference still persists on 8 x 8 pixels versus the macroscopic scale of RNA-SEQ. What evidence is that the results on the two scales are directly comparable?

209 How do the trajectories of bioluminescence on Per2 at the single cell level compare with the Per2 trajectories by transcriptomics?

We have not tracked the single cells, but we made a measurement of bioluminescent signal at the population level. For this reason, since we always analysed a population behavior rather than a single-cell one, it is possible to compare all our data together.

Transcriptomic data have the advantage of simultaneous detection of many transcripts, at the

price of a lower signal-to-noise ratio. Per2 was found to significantly oscillate only in HL condition, while other genes of the core clock were found to oscillate in all 3 conditions (HL, LH, HH), see Fig. 4g.

209 What happens to the cycles in a per2 knockout?

Thanks for this interesting comment. We aim to investigate the effects that a wrong schedule of habits (e.g. feeding and fasting) on a healthy clock. We add a phrase on the discussion about these possibilities.

269. This statement is incorrect. The authors are simply not citing related work on single cell oscillatory perturbations that have been done on other model clock systems.

Thanks for this comment, we added the most relevant papers we found (lines n. 91-96) and updated the bibliography accordingly (ref. 39-42).

312. The phase analysis is just incomplete and needs to be more thorough with an examination of phase plotted as a function of time for single cells, for example. What do the percentiles and mean phase look like over time?

We performed luminescent imaging acquisition at high spatial resolution only to set-up the entire experimental system, but we did not tract the single cells.

316. Where was this optimality result first reported?

Thanks for the suggestion, we better highlight where the optimal conditions were set.

335. It might be worth mentioning one of the author's own work and that of O'Donnell in the Royal Society on feeding and fasting and its impact on malaria as well.

Thanks for the suggestion, we added two references that can easily contextualized our work in a broader perspective.

343-6 It was mentioned that labview was used; however, the schematic that was shown did not show the full set up. A block scheme of the experimental set up maybe useful.

Thanks for this comment, we added citation of previous work and publicly provided the block scheme of the experimental set up in <https://github.com/Onelia-G/CAVRO-PUMPS-FOR-MICROFLUIDICS>.

365. Very few details are given on the microfluidic chips. What are standard soft lithography techniques? How was the mask for the device designed? What was the PDMS substrate? How were the devices pretreated? Since the 2 devices were made from PDMS, how was the autofluorescence handled? The methods section is incomplete.

Thanks for the comment, we improved the Material and Methods on the section of Microfluidic Devices and add more details in the Fig. S1.

About the autofluorescence, actually we use luminescence and not fluorescence, but even in case of luminescence PDMS does not provide any problem of autofluorescence as described in this work (The autofluorescence of plastic materials and chips measured under laser irradiation. Lab on a Chip 2005).

439. Image processing is not defined in sufficient detail to know what is being done. It would help to see what a blown up image looks like in a supplementary figure. For example, correction for cosmic ray artifacts means nothing to the reader. The image processing steps from source to detector need to be layed out. How were single cells identified? If a workflow in MATLAB was developed, it needs to put in GitHub.

We re-wrote the Image Processing in the Methods section, describing all the procedures with more details and the relative references have been added.

We obtained the video presentation by putting the images acquired every 30 min in sequence using ImageJ; cosmic ray artifacts were removed by ImageJ (function Process>Noise> Remove

Outliers) as already reported in our previous publications (Li, Y. et al. Proc. Natl. Acad. Sci. U. S. A. 117, 10350–10356, 2020; Li, Y. et al. Elife 9, 1–36, 2020).

We added our Matlab code for creating the heat map shown in Fig. 1g in GitHub and now is

publicly available at this link <https://github.com/Onelia-G/heap-map>

Comments by reviewer 1: It is still not clear from the Figure legend or text whether the plots of bioluminescence are averages over the ROIs or not.

439. Where is the single cell data deposited?

Bulk RNA-seq data presented in this study have been deposited at the Gene Expression Omnibus database (<https://www.ncbi.nlm.nih.gov/geo/query/acc.cgi?acc=GSE167763>) with the dataset identifier GSE167763 (token qvyzvuampjqrlyf).

443 Adjusting the brightness and contrast would alter the results obtained, giving false results if this was utilized during the image processing step. Actually, we erroneously wrote that we adjusted the brightness and the contrast. What we wanted to say is that we removed the cosmic ray artifacts by pixelwise comparison of two consecutive images using the minimum value of each pixel using ImageJ, as reported in the work Noise-driven cellular heterogeneity in circadian periodicity. PNAS 2020.

Comments by Reviewer 1: It is recommended that you use a comparison to luminescent beads in another chamber to get at detection noise.

445. ROI is not defined, although a region of interest could mean anything from a FOV to the region around a cell.

Thanks for the suggestion, we defined better the ROI at lines n. 494-502.

465. How was the quality of the FASTQ files assessed? Was their quality assessed using Ewels MultiQC? What are some of the quality plots graphed for different time points? Will these be added to the supplement?

The analysis of the quality of the samples was made through FastQC/MultiQC taking into account the general statistics of fastqc (Per base quality, Per sequence GC content, Overrepresented sequences). The overall report containing these information can be found here: https://drive.google.com/file/d/1IVBSN8MjsXrepH8CfsQzJrwxYX9UXoc6/view?usp=s_haring. We also addressed pre-processing quality by statistics related to alignment and quantification (amount of unique mapped reads on the total, counted reads ,detected genes), reported here: https://drive.google.com/file/d/1oiUpegdwG3aRahgST6O6ANjZ47F0DFPn/view?usp=s_haring.

642 Microfluidic device schematic representation: was the culture chamber placed on a board of some sort? What are the long serpentine areas? Is it tubing or the microfluidic device? Needs to be more specific.

Thanks for this comment, we provided more details about the microfluidic platforms used in the Material and Methods (Microfluidic Device) and also added the Figure S1.

The culture chamber is placed on a borosilicate glass slide, placed in a petri dish and located on the microscope stage, as described in the Methods section; the serpentine is a part of the microfluidic chip as the Supplementary Fig. 1 shows.

646. The authors are not presenting enough details on the microfluidic device (the schematic is ambiguous; were the devices pretreated?), and they are not making full use of its measurement capability. Why take measurements only every 6 h in Fig 1E? Why not every 30 m? If continuous observations are being taken as reported in the methods (i.e., 30

m intervals), why is the luminescence not being reported on a finer scale? We provided more details about the microfluidic platforms used in the Methods (Microfluidic Devices) and also added the Fig. S1.

Fig. 1e shows Real-Time PCR data of human fibroblasts harvested every 4 h for 48h. This technique quantifies the expression level of one or more genes of interest of the sample harvested at a precise time point. The frequency of sample collection depends on an operator who for obvious reasons cannot harvest the cells every 30 min. However, 4h interval allows to follow the temporal dynamics of a circadian gene.

Different considerations need to do for the luminescence measurement, that is based on the acquisition under the microscope of a luminescent signal expressed from a luciferase- reporter cell

line. With this approach it is possible to implement a frequency of measurement with a high temporal resolution, every 30 min, with a good quality of imaging, as we described previously; on the other hand, it is possible to follow the expression of only one gene.

647 The luminescence data goes to negative; what methods for image processing, data cleaning or manipulation was done to get the results? This was not mentioned in the image processing step. To analyze the circadian parameters, we first need to remove baseline changes because there are often drastic changes in the baseline in the first few days in culture. The baseline drift was obtained by fitting a polynomial curve by LumiCycle Software (Li, Y. et al. Epigenetic inheritance of circadian period in clonal cells. *Elife* 9, 1–36, 2020; Li, Y. et al. Noise-driven cellular heterogeneity in circadian periodicity. *Proc. Natl. Acad. Sci. U. S. A.* 117, 10350–10356, 2020; Yamazaki, S. & Takahashi, J. S. Real-time luminescence reporting of circadian gene expression in mammals. *Methods in Enzymology* 393, 288–301, 2005).

Comment by reviewer 1: Why is fitting a polynomial the right thing to do? Are there other detrending methods in the literature that have been recommended?)

679. Nonparametric test is needed.

We acknowledge that, for temporal series, data collected within the same data series are not iid (independent and identically distributed), however we applied t-test and ANOVA on parameters, like phase and frequency, to detect differences between independent experimental conditions, thus it is reasonable to use these tests.

Comments by reviewer 1: What happens when a nonparametric test is used?

Reviewer #2:

Remarks to the Author:

The authors have addressed my question accordingly. I do not have further objections.

Reviewer #3:

Remarks to the Author:

The manuscript at its current version (with the changes that the authors did and after fixing issues listed below) should be published in Nature Comm.

It's now clear why the authors used the microfluidic platform and, more importantly, the overall why/what/how of the paper.

The presented method or a similar may change how people do disease models in the future (for personalized medicine).

Before publishing, please:

- Remove one of the sentences that start at line 49 or line 52 as they provide the same information.
- Rewrite sentence starting at line 261.

General Comments by reviewer 1: Recognizing that single cell measurements are not being made is a good change, and accurately describing the microfluidic measurements over 8 x 8 pixel grid is an improvement, but it still does not remove the criticisms of reviewer 1. The questions still remain about the level of detection noise and stochastic intracellular noise on this 8 x 8 pixel scale. Approaches to addressing this problem of noise are published in the Nature family of journals on the clock. Moreover, the authors still need to cite the relevant literature in Nature Publisher family of journals and other journals where single cell measurements have been made in model systems to examine the entrainment question. What is being plotted in terms of bioluminescence is still not made clear. The phase analysis is not contemporary. For example, the phase plots in Figure 3c assume a constant phase, and this is probably not the case. This paper is not ready for publication because of shortcomings in the controls for the microfluidics experiments and the analysis of bioluminescence.

We appreciated the detailed and careful revision of the reviewer and we agree that there are some changes, especially in the data analysis that can further improve the quality of the paper. We have used a gold standard analysis in the circadian field; however, we agree that adding supplementary comparative analysis will better validate the robustness of our results. We would like to underline again that we are not proposing a single cell analysis and we are adopting all methodologies that can apply to our work, including those developed for single cell measurements in case.

Reviewer #1

The microfluidics approach is very interesting; however, the work is not tied into the clock literature, either experimental or theoretical.

We thank the reviewer for his/her appreciation, we now better contextualized our work within the literature. The following references have been added in the introduction and discussion sections:

Sato, M., Murakami, M., Node, K., Matsumura, R. & Akashi, M. The Role of the Endocrine System in Feeding-Induced Tissue-Specific Circadian Entrainment. *Cell Rep.* 8, 393–401 (2014).

Damiola, F. et al. Restricted feeding uncouples circadian oscillators in peripheral tissues from the central pacemaker in the suprachiasmatic nucleus. *Genes Dev.* 14, 2950–2961 (2000).

Stokkan, A. K. et al. Entrainment of the Circadian Clock in the Liver by Feeding Linked references are available on JSTOR for this article : Entrainment of the Circadian Clock in the Liver by Feeding. 291, 490–493 (2001).

López-Otín, C., Galluzzi, L., Freije, J. M. P., Madeo, F. & Kroemer, G. Metabolic Control of Longevity. *Cell* 166, 802–821 (2016).

Longo, V. D. & Panda, S. Fasting, Circadian Rhythms, and Time-Restricted Feeding in Healthy Lifespan. *Cell Metab.* 23, 1048–1059 (2016).

The two parts of the paper are not logically connected because one part is on the microscopic scale of single cells, and the other part is on the macroscopic scale through standard RNA transcriptomics.

Microfluidics is a quite new approach in circadian studies, only few works previously explored the potential of this technology. Given our long-term experience on different strategies to perform robust cell culture in microfluidic (for instance, Giulitti et al., Optimal periodic

perfusion strategy for robust long-term microfluidic cell culture, Lab on a chip, 2013), we carefully evaluate the potential drawbacks in using microfluidic technology for circadian study. In particular, to avoid any artifact coming from the intrinsic properties of microfluidic setup, including small volume, the high surface to volume ratio and upstream and downstream properties created by unidirectional medium flow. We thought it was important to visualize the circadian behavior in different region of the microfluidic channel and to capture any spatial heterogeneity along the channel and at the edges.

In the manuscript we had erroneously written that we performed luminescence experimental measurements with single-cell resolution. However, despite we used an imaging resolution of 20 microns, we did image analysis on a pixel basis, not cell basis. We did not perform any single-cell identification and single-cell tracking.

Thus, we now made clear this concept within the text and explain that high resolution was used for investigating the spatial heterogeneity of luminescence signals within microfluids. For avoiding misinterpretation, any reference to single-cell measurement were eliminated.

We thank the reviewer for all suggestions about the single-cell analysis, which avoids misunderstanding and, in the revised manuscript, the analysis of Per2 oscillations and the transcriptomics better connected.

We anticipate that this clarification will probably overcome many of the issues raised by the reviewer about the proper use of single-cell analysis.

Comment by reviewer 1: In that so few papers have been published on microfluidics of the clock, it would be desirable and strongly suggested that the use of microfluidics in other model systems to examine light entrainment in single cells should be cited. There are a number of papers on this topic in Nature journals and other journals that are not cited and would provide insights, such as methods including controls and analyses, on the microfluidic data presented here.

In the next answers, we will refer to the explanation given at this point.

Thanks for this comment, we revised the references and added interesting papers on microfluidics of the clock (ref. n. 42-44).

42. Deng, Z. *et al.* Synchronizing stochastic circadian oscillators in single cells of *Neurospora crassa*. *Sci. Rep.* **6**, 1–18 (2016).

43. Caranica, C., Al-Omari, A., Schüttler, H. B. & Arnold, J. Identifying a stochastic clock network with light entrainment for single cells of *Neurospora crassa*. *Sci. Rep.* **10**, 1–24 (2020).

44. Deng, Z. *et al.* Single Cells of *Neurospora Crassa* Show Circadian Oscillations, Light Entrainment, Temperature Compensation, and Phase Synchronization. *IEEE Access* **7**, 49403–49417 (2019).

The authors do not make use of the continuous time measurement at the single cell level.

As mentioned above, we agree with the reviewer that we cannot state that the measurement was performed at single-cell level, because we did not track cells. Instead, we resolved the spatial heterogeneity of luminescence signal throughout the microfluid channel surface. Our data show that a circadian oscillatory pattern can be established quite homogeneously throughout this surface, and edge effect or upstream and downstream effects are not observed.

We added this piece of information in the manuscript (lines 110-112) and clarified that we used high-resolution imaging coupled with microfluidic technology for the circadian study.

Comment by reviewer 1: If there is this no spatial homogeneity in the signal, one should be able to demonstrate this by calculating the Kuramoto K on different locations (ROIs) or by plotting phase as a function of time for different ROIs on the microfluidic device surface.

We thank the reviewer for this comment, and we decided to adopt the method suggested by him/her for the analysis.

We followed his/her suggestion and quantitatively evaluated ROIs synchrony by Kuramoto parameter, exploring different sizes of the ROIs (8-by-8, 16-by-16 and 32-by-32 pixel).

Obviously, increasing the ROI size decreases the spatial resolution. We estimated that the number of cells contained in the smallest 8-by-8 ROIs is about 20 cells, which could be considered already a bulk analysis. However, to mediate between getting a bulk cell population measurement and maintaining a high spatial resolution, we decided to analyse 16x16-pixel ROIs (ROI size of about 320x320 um, about 80 cells) in Figure 1, and added a comparative analysis based on Kuramoto order parameter in Fig. S2f.

Also, following the reviewer's suggestion, we also included plots of the instantaneous Hilbert phase as a function of time for different ROIs in Fig. S2d-e, which convey a similar message compared to Kuramoto.

Indeed, consistently to what we had already proposed by qualitative observation of Fig. 1g, higher spatial homogeneity is achieved with medium changes performed every 24 hours compared to every hour. We added these new figures and commented the results accordingly. The procedure was described in the Methods (image processing).

The description of the microfluidics needs to be more complete in Materials and Methods.

We thank the reviewer for highlighting this weakness of the manuscript. We added more details in Methods section (Microfluidic Devices- lines 392-422) and in the supplementary figure (Fig. S1) a more detailed scheme describes the microfluidic platforms used.

They do not characterize the level of detection noise in their single cell measurements, leaving in question the validity of results presented in the microfluidics experiments. The phase analysis is incomplete and not contemporary.

Also this comment falls within the single-cell measurement misunderstanding. Our main results are based on an imaging analysis on ROIs of 8x8 pixels, where the pixel size is 20 microns, according to the methods reported in Material and Methods (Image Processing). This ROI size was chosen as a trade-off between spatial heterogeneity and robustness of measurement, according to measurement noise evaluation, as pointed out by the reviewer. The signals integrated over the single pixel in the defined ROI is high and reproducible compare to the background.

These high-resolution measurements, reported in Fig. 1, provide evidence for a consistent cellular response throughout the entire surface of the culture chamber. Manuscript has been modified accordingly.

Comment by the reviewer 1: There are published accounts in Nature journals to separate the detection noise from the stochastic intracellular noise (SIN) in other clock systems at the single cell level. In that the observations depend on characterizing the level of detection noise, a similar analysis is recommended here even if the observations (detection noise+ SIN) are averaged over 8x8 pixels. What happens to the signal if the area of integration is increased?

As previously mentioned, we would like to point out that we are not measuring single-cell noise, as the ROI considered include many cells and they are representative of bulk analysis on different spatial position within microfluidic chamber.

Moreover, we would like to point out that our measurements are based on bioluminescence, which (unlike fluorescence) makes possible to drastically improve signal-to-noise ratio. We show here below an example.

The signal here has been corrected only by ImageJ image pre-processing as described in the methods of the paper, without detrending. ROIs 1-2 are background noise, ROIs 3-4 are signal from the culture chambers. ROIs 1-4 all have the same area.

We tested different ROI sizes and calculated the Kuramoto order parameter (now reported in Fig 1h and S2f). We found that 16-by-16 pixel ROIs reduce spatial heterogeneity (with $K > 0.7$), giving a more robust cell population bulk signal. Thus, now the results in the main manuscript are referring to this ROI size (see also Fig. 1g, h).

The connection to existing models of the clock is absent. For example, if the cells constitute damped oscillators, then one might argue they can entrain to any period of the driving input. On the other hand if single cells have their own intrinsic clock, they can entrain only within a range of frequencies about the intrinsic frequency of each cell. From the data presented it is not clear that single peripheral cells have a clock. Not knowing that individual cells have clocks leaves open how to explain the entrainment phenomenon observed.

This question is based on the idea that we explored the Per2 expression of the cells by a single cell measurement. However, after system validation in Figure 1 using high-spatial resolution, we performed all our experiments using a bulk analysis approach. Figure 1 will be important to assess the capability of our platform to be useful for the circadian study we have in mind, and all the important findings achieved are given by a bulk analysis of a single microfluidic channel.

However, the reviewer is asking important question if the input is able to entrain the intrinsic cell clock or not. All single cells show a cell-autonomous and self-sustained circadian oscillation with undiminished amplitude and diverse circadian periods and this circadian expression continues also during cell division as previously demonstrated (Nagoshi, E. *et al.* Circadian gene expression in individual fibroblasts: Cell-autonomous and self-sustained oscillators pass time to daughter cells. *Cell* **119**, 693–705 (2004); Welsh, D. K., Yoo, S. H., Liu, A. C., Takahashi, J. S. & Kay, S. A. Bioluminescence imaging of individual fibroblasts reveals persistent, independently phased circadian rhythms of clock gene expression. *Curr. Biol.* **14**, 2289–2295 (2004)). Cells can be partially synchronized in phase by different agents (e.g.

serum shock, dexamethasone) at the population level, however, lack of oscillator coupling in cell cultures leads to a loss of synchrony among individual cells and damping of the ensemble rhythm. Bioluminescence imaging of individual fibroblasts reveals persistent, independently phased circadian rhythms of clock gene expression.

Thus, the Per2 and transcriptomics oscillations, we observed, is always based on entrainment of the entire cellular population.

Comment by reviewer 1: With the microfluidics there is the opportunity to show that single cells can entrain to an external signal. The references above are limited to showing entrainment in bulk. It has already been demonstrated in other systems that single cells do entrain in other systems. Does it do so here at an intermediate scale of 8x8 pixels? The new results in Figure 3e-f would indicate a yes, but how does this compare with similar entrainment experiments done at the single cell in other model systems?

We do not agree with the reviewer that we can considered the ROI 8x8 as an intermediate scale. In our hand, considering the number of cells for a single ROI and considering that we are now using 16x16 to discuss the heterogeneity of our microfluidic perfusion setup, we cannot obtain any conclusion about single cell entrainment.

We also would like to stress that single-cell microfluidic studies, see references in the Introduction section, are out of the scope of this paper. Moreover, the microfluidic single-cell studies we found used droplet microfluidics to perform cell-cell separation and evaluate the signal of single cells over time. Cells used in this paper are mammalian cells that require attachment to grow and single-cell measurements would require a completely different biological system. Our acquisition of one image every 30 min has a frequency that it is still too low to perform cell tracking and bioluminescence signal needs to be integrated over longer time intervals compared to fluorescence to be robustly detected.

Line_number

43. *What is missing in the introduction is a relation of the work to other model systems on the clock, such as in Arabidopsis, where glucose entrainment was reported earlier in Nature. What is also missing is the first global descriptions of the clock network in other model systems to help interpret the transcriptomics data.*

Thanks for this comment, we revised the introduction and added this ref. n. 7 and 8 lines n.51-52.

80. *The microfluidic approach is to be commended, but their microfluidic approach to the entrainment problem is not novel and has been utilized in other model clock systems, but the authors cite none of this work. only 1 paper is missing.*

Thanks for this comment, we added the most relevant papers we found (lines n. 91-96) and updated the bibliography accordingly (ref. 39-42).

Comments by Reviewer 1: Unfortunately the most relevant papers in Nature publications and other journals in related model systems are still not being cited.

Thanks for this comment, we revised the references adding interesting papers on microfluidics of the clock (ref. n. 42-44).

42. Deng, Z. *et al.* Synchronizing stochastic circadian oscillators in single cells of *Neurospora crassa*. *Sci. Rep.* **6**, 1–18 (2016).

43. Caranica, C., Al-Omari, A., Schüttler, H. B. & Arnold, J. Identifying a stochastic clock network with light entrainment for single cells of *Neurospora crassa*. *Sci. Rep.* **10**, 1–24 (2020).

44. Deng, Z. *et al.* Single Cells of *Neurospora Crassa* Show Circadian Oscillations, Light Entrainment, Temperature Compensation, and Phase Synchronization. *IEEE Access* **7**, 49403–49417 (2019).

104. This measurement of clock gene luminescence in single cells in an entrainment experiment is not the first time this has been done. This has been done in other clock systems, but again it would be useful to relate the results here to work on other clock systems. For example, how did the authors go about separating the detection and stochastic intracellular noise in the luminescent measurements made?

We did not perform any single cell analysis. We always refer to published protocol for analysing bulk signals from cell population.

Single cell analysis would have required a different approach, as the one we published recently (Li, Y. *et al.* Noise-driven cellular heterogeneity in circadian periodicity. *Proc. Natl. Acad. Sci. U. S. A.* **117**, 10350–10356, 2020. Li, Y. *et al.* Epigenetic inheritance of circadian period in clonal cells. *Elife* **9**, 1–36, 2020).

Comments by reviewer 1: The results reported here are intermediate between single cell analysis and bulk analysis. Averaging over 8x8 pixel grid still involves a new kind of analysis that takes account of both kinds of noise, detection noise and stochastic intracellular noise, at this level. Also a relevant control of following luminescent beads in a different chamber on the same device is not being used to examine the detection noise, for example.

As explained above, to avoid further doubts we adopted 16x16 pixel ROIs which still allow analysis of heterogeneity with 320 micron resolution.

If we understood correctly, the reviewer asks us to add an additional control of a source of stable luminescence beads within the microfluidic chip. First of all, we would like to clarify that luminescence signal is generated from an enzymatic reaction in which a photon is produced by conversion of a substrate. This technique does not require any light excitation, so any photon emission will be stoichiometrically related to the presence of the catalytic enzyme and the substrate.

To address this reviewer's request, considering that we were not able to find chemiluminescence beads, we compared the signal coming from Per2::Luc fibroblasts (orange track) with luminescence (blue track) detected with a solution of peroxide solution and luminol reacting with a Horseradish peroxidase (HRP)-conjugated antibody, which was used to coated the microfluidic chamber..

From the figure here below, where we show the raw signal obtained from the CCD camera (850s CCD, Spectral Instrument), it is striking how a stable signal is detected from HRP chemiluminescence, whereas a high-amplitude oscillation is measured for Per2::Luc fibroblasts (orange track), both well-above background. This confirmed once more that Per2 oscillatory behavior is due to a biological response and not to other potential artifacts.

We decided not to include this piece of information in the manuscript though, as our biological model (Per2::Luc fibroblasts) has been widely used previously in the cited literature.

105 What fraction of the variation in luminescence is detection noise? How is the reproducibility of the measurements quantified? What controls have been done with luminescent beads substituted for living cells and the like? Without the characterization of the detection noise the microfluidic results remain in question.

We obtained the video presentation by putting the images acquired every 30 min in sequence using ImageJ; cosmic ray artifacts were removed by ImageJ (function Process>Noise> Remove Outliers) as already reported in our previous publications (Li, Y. *et al. Proc. Natl. Acad. Sci. U. S. A.* **117**, 10350–10356, 2020; Li, Y. *et al. Elife* **9**, 1–36, 2020).

We designed manually ROIs of the channels and measured the luminescence signal of each by using ImageJ (function Image>Stacks> Plot Z-axis Profile).

For the luminescence data that goes to negative, we used also LumiCycle software to subtract the baseline.

All these procedures are added in the Methods section (Image processing) and relative references have been added.

Comments by reviewer 1: The video still doesn't constitute a control or a measurement of the detection noise. What about the controls such as luminescent beads in a different chamber on the same device? What does it mean to remove cosmic ray artifacts? How is this done? This needs to be described in the paper or its supplement. There needs to be a careful analysis of the detection noise.

As mentioned before, we would like to point out that, luminescence acquisition has much less noise detected, highly quantitative reporters and suitable to measure how much and when a gene or protein is expressed than other methods (Noguchi, T. & Golden, S. *Bioluminescent and fluorescent reporters in circadian rhythm studies. BioClock Stud.* **24** (2017)).

We better detailed the description of the methods of image analysis. Briefly:

- Cosmic ray artifacts produce the so-called salt-and-pepper noise, that was removed in ImageJ by the function Process>Noise> Remove Outliers, which performs an adaptive median filtering, meaning that pixels that were detected as outliers because far above signal intensity were replaced by the median of surrounding pixels. Following the procedure previously published (Li, Y. *et al. Epigenetic inheritance of circadian period in clonal cells. Elife* **9**, 1–36 (2020); Li, Y. *et al. Noise-driven cellular*

heterogeneity in circadian periodicity. *Proc. Natl. Acad. Sci. U. S. A.* **117**, 10350–10356 (2020)).

- Then all image histograms were equalized in ImageJ using the function Image>Adjust>Brightness/Contrast>Auto. This step was performed after arranging the images in stack to ensure the equalization procedure was applied in the same way to each image and preserving comparability.
- Images were then exported in 16-bit .tif format from ImageJ.
- LumiCycle software (Actimetrics Inc., Evanston, IL) was used to subtract the baseline, estimated by a running average of the data. We recognize that many algorithms have been developed for background subtraction, the one used in LumiCycle is quite established in the field for treating luminescence images. However, given that Lumicycle is a bit a black box in this respect, we performed a moving average detrending as now described in the Methods of the paper for initial validation of the robustness of our data analysis pipeline.

All this information was added in the methods, in Imaging Processing section.

107 *The authors have not taken advantage of this system and shown continuous data acquisition. Why have they not done so, if the systems is so reproducible? How do the findings on Per2::Luc compare with homologs in other systems, where single cell measurements have been taken on the clock mechanism?*

In view of guaranteeing reproducibility, the continuous acquisition of luminescence signal was integrated over 30-minute intervals to increase signal-to-noise ratio. The phrase n. 107 has been removed and the details are better explained.

Comments by reviewer 1. The homolog comparison done at the single cell level in another model system has not been done.

We agree with the reviewer that it would be interesting to reproduce our results in a model system, but this is out of the scope of this work and presents the challenges we described above.

127. *The authors are not using reasonable and well studied measures of phase, such as the Hilbert phase. These contemporary phase measures like Hilbert phase allow one to get away from assuming constant phase with the phase shift. These Hilbert phase measures are designed to deal with the case of phase changing with time as in Fig 4e.*

Considering the line 127, and the Fig. S2c, we measured the phase shift as time difference between two corresponding peaks of the two media changes implemented as schematized in Fig. S2a.

We agree with the reviewer that more accurate estimations of the phase exist. However, RNA-seq data were acquired for the time span of a single cycle. The phase was determined using the quite established algorithm JTK. This phase determination is just a rough estimation: as shown in Fig. 4e, the phase is determined with 2h time resolution. Our biological conclusions are based on large phase differences. We agree that we could have missed more subtle transcriptional differences, but such results would also require a different experimental design with RNA sampling for a much longer period of time.

Comments by the reviewer 1: There are better ways to do the phase analysis. As an example the polar plot Figure 3c assumes constant phase, which is unlikely to hold in an entrainment experiment.

The polar plot displayed in Fig. 3c shows the time-deviation of the acrophase (the time at which the peak of a profile occurs) at a defined time point (Day 2 of Free Running). Thus, it is not making the assumption of constant period/phase shift. We improved the figure caption to make clearer what it is represented.

Considering the reviewer's interesting comment regarding the phase analysis, we also added a new analysis using instantaneous Hilbert phase over time as reported in Fig. S5a.

On the contrary, for RNA-seq data we decided to still rely on the commonly used JTK algorithm, especially considering that we have measurements for a single-period timespan.

134 Synchronization is not quantified. What kind of synchronization? Is it phase synchronization? Is it period locking?

Our study is focused on how repeated stimulations of feeding and fasting cycles are able to synchronize the phase of Per2 of the cell population. Analysis of the period was initially excluded from the analysis, excepted for the first characterization in the microfluidics use as shown in Fig. S2b. However, we did an additional experiment reported in Fig. 3e-f to better understand if our stimulus of glucose/insulin applied at different cycle times (20h, 24h, 26h) provides free-running rhythms, or zeitgeber-dependent rhythms. We found that the stimulation of feeding (HIGH) and fasting (LOW) cycles of 20h and 26h significantly change the period of Per2 expression compare to 24h cycles.

Comments and discussion about this new experiment have been added.

Comment by Reviewer 1: Thank you for the additional experiment. It would be very nice to overlay the driver in 3a-b on 3-d. Also a plot of phase of bioluminescence for LH-1, LH-2, and LH-3 as a function of time on the same graph would be good. This would help address how things are changing with regard to phase. Phase may be changing with time also, but this is not captured in the analysis. Also only by choosing the appropriate phase measure can you independently test for a period change and a phase change.

Considering reviewer's comment, we decided to add a new analysis of instantaneous Hilbert phase both for HL-1,2,3 and LH-1,2,3 and cyclic HH and HL imposed at different at different time windows (10:10h, 12:12h, 13:13h), as shown in Fig. S5. All together the results showed that the phase is constant and the flow of the data supports the main message of the paper. Indeed, we added where required new information but without changing the order of the figures.

171. How is alignment or misalignment quantified? How do the authors quantify the synchronization in Per2 under LH-1, LH-2, and LH-3? What is the phase variation in each experiment over time? The authors have stated they have single cell resolution. What are the percentiles (across cells) in the phase over time?

The alignment and the misalignment depend on the coherence between the time of when we stimulate the cells, that always corresponds to the trough of Per2 expression and the type of the stimulus applied, HIGH (feeding) or LOW (starvation). We better specified this concept in the manuscript.

Considering that the activity phase (when glucose and insulin are high) occurs between the minimum and the maximum of Per2, and the rest activity (when glucose and insulin are low) occurs between the maximum and the minimum of Per2, HL mimics an aligned condition since the HIGH Stimulus is applied coherently with the phase activity. On the other hand, LH mimics a misaligned condition since the LOW Stimulus is applied incoherently with the phase activity. The phase variation is the time difference between two corresponding peaks of the two

experimental conditions compared as we explained previously. The variability of each experiment has now been evaluated and reported in the captions.

As already explained in the previous questions, we did not do any single cell resolution analysis but only bulk.

Comments by reviewer 1: It is possible to measure the metabolites (glucose) simultaneously in these living cells now and needs to be tracked along with Per2.

Glucose concentration measurement by fluorescence probes would be complicated to perform in parallel with luminescence acquisition in our experimental system, because luminescence does not require laser excitation like fluorescence, thus our system only includes the detector. However, even overcoming the technical difficulties, fluorescent glucose analogs have been shown not to be transported and intracellularly metabolized with the same kinetics of glucose (Zambon, A. *et al.* High Temporal Resolution Detection of Patient-Specific Glucose Uptake from Human ex Vivo Adipose Tissue On-Chip. *Anal. Chem.* **87**, 6535–6543 (2015).), and what is critical for understanding metabolic activity is the flux of metabolized glucose, which cannot be derived from the knowledge of the glucose concentration measurements only, as we previously showed (Zambon, A. *et al.* High Temporal Resolution Detection of Patient-Specific Glucose Uptake from Human ex Vivo Adipose Tissue On-Chip. *Anal. Chem.* **87**, 6535–6543 (2015).).

162. If the aim is to show entrainment, why are not oscillatory patterns to glucose and insulin being applied rather than a single pulse? Setting up a repeated pulse would be more informative and show true entrainment to the glucose signal. It might be useful to mention at this point that the periodic pulse is imposed later, and this whole setup is a control.

The experiment reported in Fig. 2a-b aimed at deciding the time and the frequency at which we switch between H (feeding) and L (fasting) media and vice versa within a 24h cycle, among these combinations: 4:20h, 8:16h, 12h:12h, 16:8h and 20:4h. Once we decided that 12h:12h better sustains the Per2 oscillations, we did repeated HIGH and LOW stimulations up to 3 days.

We rephrase the related text to make clearer the idea that only aligned 24-hour HL cycles are able to synchronize the cell population and provides strong entrainment.

165. What does the periodic signal look like when the power spectra are computed on single cells using the 30 m resolution data? Fig 1B is not sufficient. How do the authors handle the cells settling into a stable limit cycle? Why is the luminescent response damping out? What experiments have been done to explain the damping?

As mentioned above, reported data are not single-cell data, and the subsequent questions are still related to this aspect. Signal damping is due to desynchronization as previously reported, not to single-cell damping.

Indeed, during the Free Running acquisition, the luminescence damping occurs because of progressive phase desynchronization between cells belonging to the whole culture (Nagoshi *et al.*, 2004; Welsh *et al.*, 2004). While single-cell luciferase emission in fibroblasts remains high throughout and self-sustained (Nagoshi *et al.*, 2004; Welsh *et al.*, 2004, Li Y. *et al* 2020), the whole culture undergoes toward a desynchronization due to the absence of intercellular signaling coupled arising for example from cell division (Nagoshi *et al.*, 2004).

On the other hand, what we observed in our experiments is that 12/12 HL (for 1, 2 3 cycles) matches the phase of the Per2 oscillation and no damping was observed, as it is possible to appreciate in Fig. 3a-b.

About the Figure 1b, we added Figure S1 to increase the details of the microfluidic platforms used.

Comment by Reviewer 1: Even doing in bulk does not eliminate the problem of settling into a stable limit cycle. That problem persists at the macroscopic level as well or an intermediate level such as 8 x 8 pixel level.

As we can see from the temporal profiles, at the bulk level of the (now) 16-by-16 pixel ROIs cells are not settling into a stable limit cycle. This has been observed also in bulk measurements after dexamethasone stimulation, and we cannot say from these data if it is a matter of entrainment only or if the single-cell clock is not stable.

167. Why does the 12:12 give the most sustained pulse? How does this relate to the early work of Pittendrigh and other findings at the single cell level in clock systems?

We analysed the overall cell population, in which the fibroblast cell period is heterogeneous, the most common period of *Per2* is between 24 and 25h (Tanya L. Leise, Persistent Cell-Autonomous Circadian Oscillations in Fibroblasts Revealed by Six-Week Single-Cell Imaging of PER2::LUC Bioluminescence, Plos one 2012; David K. Welsh, Bioluminescence Imaging of Individual Fibroblasts Reveals Persistent, Independently Phased Circadian Rhythms of Clock Gene Expression, Current Biology 2004). Our data showed that using 12/12 HL and 24h cycle, we were able to match the intrinsic clock of the cells and obtain sustained signals.

We discussed that the interplay between the frequency of the metabolic entrainment and the peripheral circadian clock need to be aligned.

The effectiveness of 12:12 may indeed relate to Pittendrigh's hypothesis of "resonance" during entrainment in which the amplitude of the circadian oscillator is enhanced when the driving entrainment signal and the underlying circadian clock are in resonance (i.e., have similar period values).

Although deeply understanding of why the resonance hypothesis seems to be interesting, the study of specific molecular mechanisms is beyond the aim of this study.

206. There is a conceptual gap here. The transcriptomic results are being done on a macroscopic scale and being compared with single cell results. Why do the authors expect to see a correspondence? Why do the authors expect the microscopic results to recapitulate the macroscopic results? What happens when single cell transcriptomics are performed?

This comment, as the others, is related to the single-cell circadian analysis. We performed luminescent imaging acquisition with high spatial resolution to set-up the entire experimental microfluidic system, but we did not track the single cells. Our data are always analysed at population level.

Comment by Reviewer 1: The scale difference still persists on 8 x 8 pixels versus the macroscopic scale of RNA-SEQ. What evidence is that the results on the two scales are directly comparable?

Our new analyses of Kuramoto order parameter at different ROI sizes support the observation that the chamber size (which is the cell population size measured in RNA-seq analysis) has a macroscopic dynamic behavior similar to the 16-by-16 ROIs, as for *Per2* reporter gene. This can be appreciated also qualitatively by the following figures, which can be compared with the above dynamics in single ROIs.

209 How do the trajectories of bioluminescence on *Per2* at the single cell level compare with the *Per2* trajectories by transcriptomics?

We have not tracked the single cells, but we made a measurement of bioluminescent signal at the population level. For this reason, since we always analysed a population behavior rather than a single-cell one, it is possible to compare all our data together.

Transcriptomic data have the advantage of simultaneous detection of many transcripts, at the price of a lower signal-to-noise ratio. *Per2* was found to significantly oscillate only in HL condition, while other genes of the core clock were found to oscillate in all 3 conditions (HL, LH, HH), see Fig. 4g.

209 What happens to the cycles in a *per2* knockout?

Thanks for this interesting comment. We aim to investigate the effects that a wrong schedule of habits (e.g. feeding and fasting) on a healthy clock. We add a phrase on the discussion about these possibilities.

269. This statement is incorrect. The authors are simply not citing related work on single cell oscillatory perturbations that have been done on other model clock systems.

Thanks for this comment, we added the most relevant papers we found (lines n. 91-96) and updated the bibliography accordingly (ref. 39-42).

312. The phase analysis is just incomplete and needs to be more thorough with an examination of phase plotted as a function of time for single cells, for example. What do the percentiles and mean phase look like over time?

We performed luminescent imaging acquisition at high spatial resolution only to set-up the entire experimental system, but we did not tract the single cells.

316. Where was this optimality result first reported?

Thanks for the suggestion, we better highlight where the optimal conditions were set.

335. It might be worth mentioning one of the author's own work and that of O'Donnell in the Royal Society on feeding and fasting and its impact on malaria as well.

Thanks for the suggestion, we added two references that can easily contextualized our work in a broader perspective.

343-6 It was mentioned that labview was used; however, the schematic that was shown did not show the full set up. A block scheme of the experimental set up maybe useful.

Thanks for this comment, we added citation of previous work and publicly provided the block

scheme of the experimental set up in <https://github.com/Onelia-G/CAVRO-PUMPS-FOR-MICROFLUIDICS>.

365. Very few details are given on the microfluidic chips. What are standard soft lithography techniques? How was the mask for the device designed? What was the PDMS substrate? How were the devices pretreated? Since the 2 devices were made from PDMS, how was the autofluorescence handled? The methods section is incomplete.

Thanks for the comment, we improved the Material and Methods on the section of Microfluidic Devices and add more details in the Fig. S1. About the autofluorescence, actually we use luminescence and not fluorescence, but even in case of luminescence PDMS does not provide any problem of autofluorescence as described in this work (The autofluorescence of plastic materials and chips measured under laser irradiation. Lab on a Chip 2005).

439. Image processing is not defined in sufficient detail to know what is being done. It would help to see what a blown up image looks like in a supplementary figure. For example, correction for cosmic ray artifacts means nothing to the reader. The image processing steps from source to detector need to be layed out. How were single cells identified? If a workflow in MATLAB was developed, it needs to put in GitHub.

We re-wrote the Image Processing in the Methods section, describing all the procedures with more details and the relative references have been added. We obtained the video presentation by putting the images acquired every 30 min in sequence using ImageJ; cosmic ray artifacts were removed by ImageJ (function Process>Noise> Remove Outliers) as already reported in our previous publications (Li, Y. *et al. Proc. Natl. Acad. Sci. U. S. A.* **117**, 10350–10356, 2020; Li, Y. *et al. Elife* **9**, 1–36, 2020). We added our Matlab code for creating the heat map shown in Fig. 1g in GitHub and now is publicly available at this link <https://github.com/Onelia-G/heap-map>

Comments by reviewer 1: It is still not clear from the Figure legend or text whether the plots of bioluminescence are averages over the ROIs or not.

Regarding the imaging processing, we increased the quantity of details in both Figure legends and methods. We thank the reviewer for this suggestion.

439. Where is the single cell data deposited?

Bulk RNA-seq data presented in this study have been deposited at the Gene Expression Omnibus database (<https://www.ncbi.nlm.nih.gov/geo/query/acc.cgi?acc=GSE167763>) with the dataset identifier GSE167763 (token qvzvwuampjqrlyf).

443 Adjusting the brightness and contrast would alter the results obtained, giving false results if this was utilized during the image processing step.

Actually, we erroneously wrote that we adjusted the brightness and the contrast. What we wanted to say is that we removed the cosmic ray artifacts by pixelwise comparison of two consecutive images using the minimum value of each pixel using ImageJ, as reported in the work Noise-driven cellular heterogeneity in circadian periodicity. PNAS 2020.

Comments by Reviewer 1: It is recommended that you use a comparison to luminescent beads in another chamber to get at detection noise.

We already addressed this question above.

445. ROI is not defined, although a region of interest could mean anything from a FOV to the region around a cell.

Thanks for the suggestion, we defined better the ROI at lines n. 494-502.

465. How was the quality of the FASTQ files assessed? Was their quality assessed using Ewels MultiQC? What are some of the quality plots graphed for different time points? Will these be added to the supplement?

The analysis of the quality of the samples was made through FastQC/MultiQC taking into account the general statistics of fastqc (Per base quality, Per sequence GC content, Overrepresented sequences). The overall report containing these information can be found here: <https://drive.google.com/file/d/1IVBSN8MjsXrepH8CfsQzJrwxYX9UXoc6/view?usp=sharing>.

We also addressed pre-processing quality by statistics related to alignment and quantification (amount of unique mapped reads on the total, counted reads, detected genes), reported here: <https://drive.google.com/file/d/1oiUpegdwG3aRahgST6O6ANjZ47F0DFPn/view?usp=sharing>.

642 Microfluidic device schematic representation: was the culture chamber placed on a board of some sort? What are the long serpentine areas? Is it tubing or the microfluidic device? Needs to be more specific.

Thanks for this comment, we provided more details about the microfluidic platforms used in the Material and Methods (Microfluidic Device) and also added the Figure S1.

The culture chamber is placed on a borosilicate glass slide, placed in a petri dish and located on the microscope stage, as described in the Methods section; the serpentine is a part of the microfluidic chip as the Supplementary Fig. 1 shows.

646. The authors are not presenting enough details on the microfluidic device (the schematic is ambiguous; were the devices pretreated?), and they are not making full use of its measurement capability. Why take measurements only every 6 h in Fig 1E? Why not every 30 m? If continuous observations are being taken as reported in the methods (i.e., 30 m intervals), why is the luminescence not being reported on a finer scale?

We provided more details about the microfluidic platforms used in the Methods (Microfluidic Devices) and also added the Fig. S1.

Fig. 1e shows Real-Time PCR data of human fibroblasts harvested every 4 h for 48h. This technique quantifies the expression level of one or more genes of interest of the sample harvested at a precise time point. The frequency of sample collection depends on an operator who for obvious reasons cannot harvest the cells every 30 min. However, 4h interval allows to follow the temporal dynamics of a circadian gene.

Different considerations need to do for the luminescence measurement, that is based on the acquisition under the microscope of a luminescent signal expressed from a luciferase-reporter cell line. With this approach it is possible to implement a frequency of measurement with a high temporal resolution, every 30 min, with a good quality of imaging, as we described previously; on the other hand, it possible to follow the expression of only one gene.

647 The luminescence data goes to negative; what methods for image processing, data cleaning or manipulation was done to get the results? This was not mentioned in the image processing step.

To analyze the circadian parameters, we first need to remove baseline changes because there are often drastic changes in the baseline in the first few days in culture. The baseline drift was obtained by fitting a polynomial curve by LumiCycle Software (Li, Y. *et al.* Epigenetic inheritance of circadian period in clonal cells. *Elife* **9**, 1–36, 2020; Li, Y. *et al.* Noise-driven cellular heterogeneity in circadian periodicity. *Proc. Natl. Acad. Sci. U. S. A.* **117**, 10350–10356, 2020; Yamazaki, S. & Takahashi, J. S. Real-time luminescence reporting of circadian gene expression in mammals. *Methods in Enzymology* **393**, 288–301, 2005).

Comment by reviewer 1: Why is fitting a polynomial the right thing to do? Are there other detrending methods in the literature that have been recommended?)

As explained above, we agree with the reviewer that there are many methods to do baseline subtraction, all with their pros and cons. Following reviewer's suggestion, we included in the manuscript a different method (centered moving average detrending) to demonstrate our results are robust to the choice of the detrending algorithm.

679. Nonparametric test is needed.

We acknowledge that, for temporal series, data collected within the same data series are not iid (independent and identically distributed), however we applied t-test and ANOVA on parameters, like phase and frequency, to detect differences between independent experimental conditions, thus it is reasonable to use these tests.

Comments by reviewer 1: What happens when a nonparametric test is used?

We analyzed the data also Wilcoxon-Mann-Whitney test, and we found significant differences with a p -value < 0.05 for any pair of conditions. We preferred to maintain the original figures and statistical analysis based on t-test and one-way ANOVA previously described.

Reviewer #2 (Remarks to the Author):

The authors have addressed my question accordingly.

I do not have further objections.

Thanks for appreciating our revision work.

Reviewer #3 (Remarks to the Author):

The manuscript at its current version (with the changes that the authors did and after fixing issues listed below) should be published in Nature Comm. It's now clear why the authors used the microfluidic platform and, more importantly, the overall why/what/how of the paper.

The presented method or a similar may change how people do disease models in the future (for personalized medicine).

Before publishing, please:

- Remove one of the sentences that start at line 49 or line 52 as they provide the same information.
- Rewrite sentence starting at line 261.

Thanks for this observation, we removed the sentences from line 49 to line 52 and rewrote the sentence starting at line 261.

Reviewers' Comments:

Reviewer #1:

Remarks to the Author:

Review of "Synchronization between peripheral circadian clock and feeding-fasting cycles in microfluidic device sustains oscillatory pattern of transcriptome"

Summary: I am happy with this revision. I am signing off. At some point they will get the proper control with the luminescent beads in the microfluidics device; alternatively they can use a dual labeling system to partition the noise into measurement error and stochastic intracellular noise, provide the labels are not overlapping in spectra.

Specific comments are referenced to the page and line number p.l

p.l

11.251 nice result with instantaneous phase. The maximum signal at the natural period is reminiscent of the Pittendrigh compensation hypothesis as noted on 14.355.

12.267. Again very interesting. The organism has different metabolic programs it can run. It will be interesting to see what these programs are.

Author seems to have addressed most of the questions, looks good to be published with just a few minor corrections.

p.7 Interesting to see how Kuramoto K is affected by the frequency of medium delivery.

p.12.287 Circadian Rhythms should not be capitalized.

p.39 Figure 1(g) and (h) are missing a scale bar.

Supplementary Figure 5 text summary "rhat" should be "that".

ANSWERS TO REVIEWER #1 COMMENTS

Specific comments are referenced to the page and line number p.l

p.l

11.251 nice result with instantaneous phase. The maximum signal at the natural period is reminiscent of the Pittendrigh compensation hypothesis as noted on 14.355.

12.267. Again very interesting. The organism has different metabolic programs it can run. It will be interesting to see what these programs are.

Author seems to have addressed most of the questions, looks good to be published with just a few minor corrections.

p.7 Interesting to see how Kuramoto K is affected by the frequency of medium delivery.

Thanks for appreciating our revision work.

p.12.287 Circadian Rhythms should not be capitalized.

We revised the manuscript, as suggested.

p.39 Figure 1(g) and (h) are missing a scale bar.

We revised the 2 figures adding the scale bar, as suggested.

Supplementary Figure 5 text summary “rhat” should be “that”.

We revised the caption of Figure S5.